# Proteosomal degradation of NSD2 by BRCA1 promotes leukemia cell differentiation

Jin Woo Park[1], Joo-Young Kang[1], Ja Young Hahm[1], Hyun Jeong Kim[1] & Sang-Beom Seo [1✉]

The human myelogenous leukemic cell line, K562 undergoes erythroid differentiation by exposure to hemin. Here, we uncovered NSD2 as an innate erythroid differentiation-related factor through a genome-wide CRISPR library screen and explored the regulatory role of *NSD2* during myeloid leukemia cell differentiation. We found that NSD2 stability was disrupted by poly-ubiquitination in differentiated K562 cells. Proteomic analysis revealed an interaction between NSD2 and an E3 ubiquitin ligase, BRCA1, which ubiquitylates NSD on K292. Depletion of BRCA1 stabilized NSD2 protein and suppressed K562 cell differentiation. Furthermore, BRCA1 protein level was decreased in bone marrow tumor, while NSD2 level was elevated. Surprisingly, among BRCA1 mutation(s) discovered in lymphoma patients, BRCA1 K1183R prevented its translocation into the nucleus, failed to reduce NSD2 protein levels in hemin-treated K562 cells and eventually disrupted cell differentiation. Our results indicate the regulation of NSD2 stability by BRCA1-mediated ubiquitination as a potential therapeutic target process in multiple myeloma.

[1] Department of Life Science, College of Natural Sciences, Chung-Ang University, Seoul 156-756, Republic of Korea. ✉email: sangbs@cau.ac.kr

Functional genomic screening has been developed by application of the gene-specific editing technology including RNAi, shRNA, and CRISPR-Cas9[1,2]. Recently, these screens have introduced variety of cellular pathways and potential drug targets. Previously, RNA interference (RNAi) or short hairpin RNA (shRNA) libraries were typically used for identifying the mechanism. However, these screening have various problems such as varying efficiencies and off-target effects of RNAi/shRNAs[3]. CRISPR-Cas9-based functional genomics was able to overcome limitations of RNAi/shRNA library methods and prevented uncertain effects of residual low-level protein expression that remains after knockdown[4].

Multiple Myeloma SET domain (MMSET/NSD2) is overexpressed in 20% of multiple myeloma patients with translocation (4; 14), the gene implicated in oncogenesis[5]. NSD2 isoforms MMSET I, MMSET II, and RE-IIBP are involved in several molecular processes, such as transcriptional regulation, DNA repair, and RNA processing[6,7]. NSD2 is also overexpressed in many different types of cancers, including gastrointestinal carcinoma, lung carcinoma, and leukemia[8]. As a histone methyltransferase, NSD2 has specific lysine residue H3K36; other sites such as H3K27 and H4K20 are found in vitro but not in vivo[7,9]. NSD2 also methylates non-histone proteins including AURKA and PTEN, which regulate cellular sensitivity to DNA damage and thus increase cancer proliferation[10,11]. Knockout (KO) of NSD2 induces Wolf-Hirschhorn syndrome, where lymphocytes show abnormal functions, including deficiency in antibody production[12]. In addition, retinoic acid-mediated differentiation of human neuroblastoma cells leads to a dramatic downregulation of NSD2[13]. Although studies have shown the involvement of NSD2 in hematopoietic cell differentiation[14], its fine molecular regulation mechanism remains largely elusive.

Breast cancer 1 (BRCA1) is a tumor suppressor gene and has an E3 ubiquitin ligase activity which is involved in DNA repair, cell cycle and transcriptional regulation[15]. Low expression of BRCA1 is correlated with breast cancer transformation and the risk of developing leukemia was increased in patients who received treatment for BRCA1 mutation-associated breast cancer[16–18]. BRCA1 increases ubiquitination of H2A, H2AX, and FANCD2 to make a binding-platform for signaling proteins related in DNA repair and differentiation maintenance[19,20].

In this study, we used a large-scale CRISPR genetic knockout screening approach to investigate the mechanisms of K562 cell differentiation to erythroid. We identified multiple targetable genes whose depletion regulate hematopoietic differentiation such as NSD2. NSD2 protein level decreased during K562 cell differentiation. In microarray analysis, NSD2 regulated target genes, which are especially related to differentiation of leukemic cell to erythrocyte. Proteomic data analysis revealed interaction between NSD2 and BRCA1 and further study revealed that BRCA1 ubiquitinates NSD2 at K292 residue. Consequently, knockdown of BRCA1 prevented K562 cell differentiation via maintaining NSD2 stability. Moreover, mutation of BRCA1 at K1183, which are regularly identified during lymphoma, disrupted erythrocyte differentiation by inhibiting NSD2-BRCA1 interaction in hemin-mediated K562 cell differentiation.

## Results

**NSD2 disrupts hemin-mediated K562 cell differentiation**. Differentiation cancer cell therapy forces malignant cells to undergo terminal differentiation[21]. However, relatively little is known about the differentiation mechanism of erythroleukemic cell line K562. To obtain better understanding of this mechanism, we carried out a genome-wide CRISPR-Cas9 mutagenesis screening (encompassing 76,441 single guide RNAs (sgRNAs)[22]) to identify mutants of K562

cells affecting hemin-mediated erythroid differentiation (Fig. 1a). We performed lentiviral infection of sgRNA library in Cas9-expressing K562. Cells were treated with NaOH or hemin for 3 days, and we sorted CD235A-positive K562 cells which indicates erythroid differentiation. We compared the screening data obtained from either NaOH-treated K562 cells or differentiated K562 cells using CRISPR-Cas9 guide score, which is the median-corrected log fold change in abundance of each sgRNA[23]. Next, we performed ontology analysis based on the gene lists that is likely to have effects on differentiation (shRNA fold change [FC] >2) using DAVID ontology tool, and identified MAPK cascade, granulocyte differentiation and hematopoietic progenitor cell differentiation (Supplementary Fig. 1A). For further analysis, we examined genes whose at least three sgRNAs were exclusively increased in CD235A-positive K562 cells versus NaOH-treated K562 cells (Supplementary Fig. 1B). Using this screening strategy, we were able to identify several genes which are known to be critical for hematopoietic progenitor cell differentiation including RUNX1, SOX4, and DOCK1[24–26] (Fig. 1b and Supplementary Fig. 1B). Next, we designed four sgRNAs targeting each gene such as PAX1, SIRT3, SOX4, and NSD2 which are identified CRISPR screening. Since we used CD235A as a differentiation marker, we tested whether each genes regulates CD235A mRNA levels. Knockout of these genes did not affect CD235A expression levels (Supplementary Fig. 1C). Depletion of these genes induced expressions of erythroid differentiation marker, globin genes such as HBE1 and HBA1/2 during hemin-mediated differentiation (Fig. 1c) and accelerated reduction of GATA1 expression by hemin (Supplementary Fig. 1D). O-dianisidine-positive cell counts increased during hemin-mediated differentiation. Knockout of these genes increased o-dianisidine-positive cell counts more than control cells (Fig. 1d). These results suggested that depletion of NSD2 had strong effects on K562 cell differentiation. For further understanding the role of NSD2 during hemin-mediated erythroid differentiation, we generated a stable NSD2 WT or Y1118A, HMTase deficient mutant, overexpressing cell line by lentiviral infection in the human erythroleukemic cell, K562 (Supplementary Fig. 2A, B). Treatment of hemin upregulated a globin gene expression levels; however, overexpression of NSD2 disrupted hemin-mediated increase of HBE1 and HBA1/2 levels. Surprisingly, NSD2 Y1118A, HMTase deficient mutant, did not affect K562 cell differentiation (Fig. 1e). Although o-dianisidine-positive cell counts increased during hemin-mediated differentiation, overexpression of NSD2 did not result in profound phenotypic changes in hemin-treated K562 cells. However, NSD2 Y1118A showed a little increase in o-dianisidine-positive cell count (Fig. 1f and Supplementary Fig. 1E). Consistent with our CRISPR-Cas9 screening, NSD2 knockdown increased o-dianisidine-positive cell counts further in hemin-treated K562 cells than that of control cells (Supplementary Fig. 2C–F). Furthermore, FACS analysis showed that overexpression of NSD2 inhibited increase of hemin-mediated differentiation marker CD235A, while NSD2 Y1118A was slightly less effective (Fig. 1g). Since the enhanced erythroid differentiation was accompanied by a marked reduction of proliferation[27], we investigated whether dysregulation of NSD2 affects K562 cell proliferation. Cell proliferation assay using NSD2-depleted and GFP positive K562 cell demonstrated that depletion of NSD2 reduced cell proliferation (Supplementary Fig. 2C, G). FACS analysis showed that knockdown of NSD2 increased apoptotic cells (Supplementary Fig. 2H). Moreover, western blotting detected cleavage of apoptosis marker, PARP (Supplementary Fig. 2I), suggesting that knockdown of NSD2 disrupts cell proliferation via induction of apoptosis and erythrocyte differentiation.

**NSD2 expression was decreased by polyubiquitination of NSD2**. Having shown that NSD2 inhibited cell differentiation

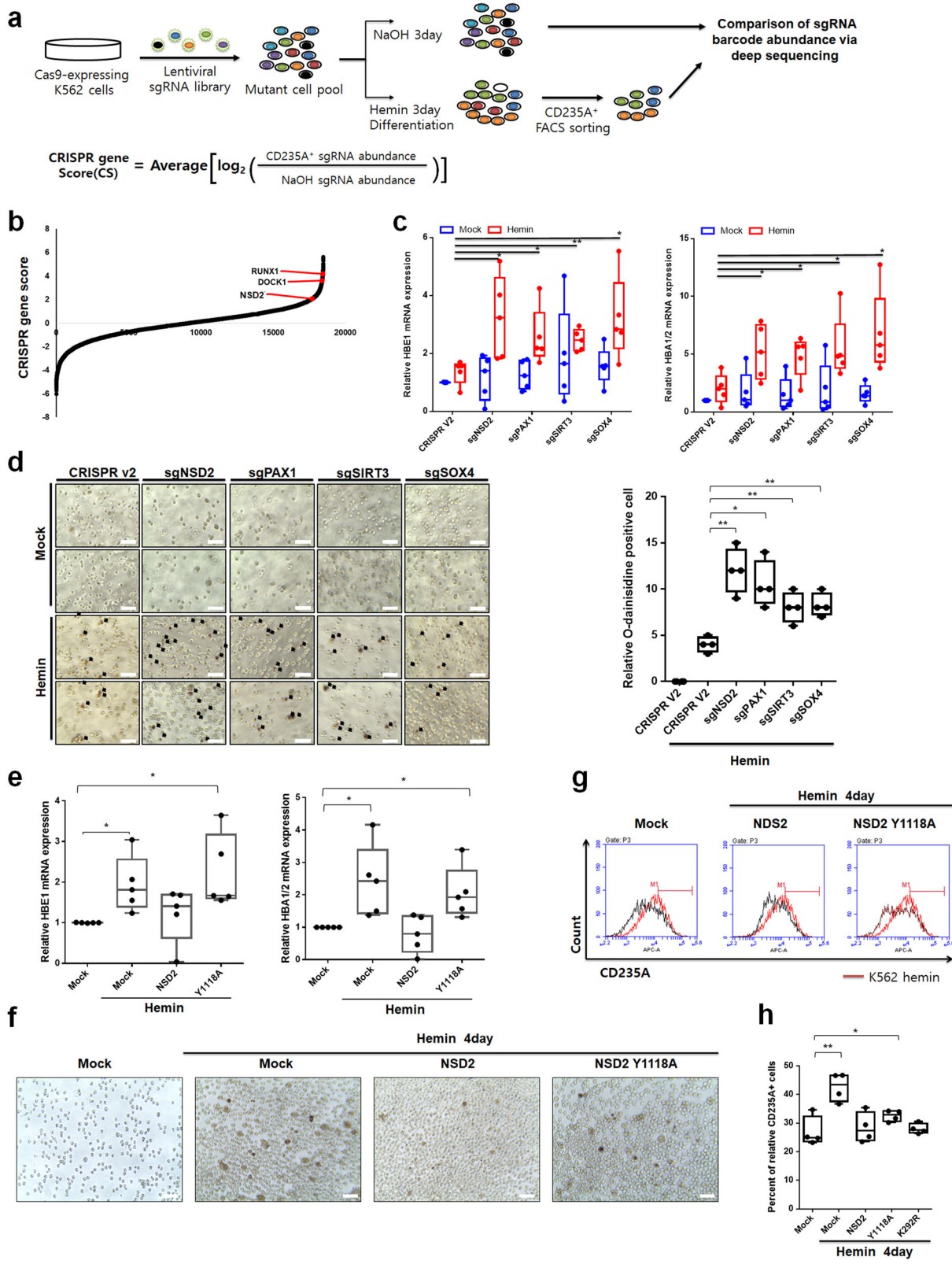

$$\text{CRISPR gene Score(CS)} = \text{Average}\left[\log_2\left(\frac{\text{CD235A}^+ \text{ sgRNA abundance}}{\text{NaOH sgRNA abundance}}\right)\right]$$

into erythrocytes (Fig. 1), we tested *NSD2* expression level in differentiated K562 cells. Interestingly, NSD2 protein level decreased during hemin-mediated differentiation (Fig. 2a, left), but mRNA levels remained unaltered (Fig. 2a, right). Since NSD2 protein level was decreased, we hypothesized that the protein level of NSD2 might be regulated by proteasomal degradation during hemin-mediated differentiation. When treated with a proteasome inhibitor, MG132, hemin-mediated reduction of NSD2 did not appear in K562 cells (Fig. 2b). We confirmed sodium butyrate (NaBu), erythroid differentiation inducer, induced K562 cell differentiation through decrease of GATA1 (Supplementary Fig. 3A). Consistent with previous data, protein

**Fig. 1 NSD2 disrupts K562 cell differentiation to erythroid. a** Schematic representation of genome-wide human knockout screen in K562 cell during hemin-mediated differentiation. **b** CRISPR-Cas9 guide rank score (derived from average of the four shRNA CRISPR-Cas9 guide scores within sgRNAs targeting a gene) for NaOH or erythroid differentiation positive cells by hemin selected after 3 days in culture. **c** The levels of *HBE1* and *HBA1/2* mRNA in K562 cells, depleting NSD2, PAX1, SIRT3, or SOX4 and treated with 30 μM hemin, were quantified using real-time PCR. Results are presented as mean ± SEM, $n = 5$; *$p < 0.05$ and **$p < 0.01$. **d** Cell differentiation was measured by staining K562 cells depleting NSD2, PAX1, SIRT3, or SOX4 with o-dianisidine (left). Cells stained in brown (black arrow indicate o-dianisidine-positive cells) indicate hemoglobin accumulation. Scale bars, 25 μm. Quantification was shown (right). Results are presented as mean ± SEM, $n = 4$; *$p < 0.05$ and **$p < 0.01$. **e** The levels of *HBE1* and *HBA1/2* mRNA in K562 cells, overexpressing WT or Y1118A variant of NSD2 and treated with 30 μM hemin, were quantified using real-time PCR. Results are presented as mean ± SEM, $n = 5$; *$p < 0.05$ and **$p < 0.01$. **f** Cell differentiation was measured by staining K562 cells overexpressing WT or Y1118A variant of NSD2 with o-dianisidine. Cells stained in brown (black arrow indicate o-dianisidine-positive cells) indicate hemoglobin accumulation. Scale bars, 25 μm. **g** Induction of cell differentiation by hemin was measured by staining K562 cells overexpressing WT or Y1118A variant of NSD2 with anti-CD235A for 1 h and sorting by FACS. The value of hemin-treated K562 cells (red line) was used as a control. **h** Quantification of FACS data in K562 cells overexpressing WT, Y1118A or K292R variant of NSD2 with anti-CD235A. Results are presented as mean ± SEM, $n = 4$; **$p < 0.01$.

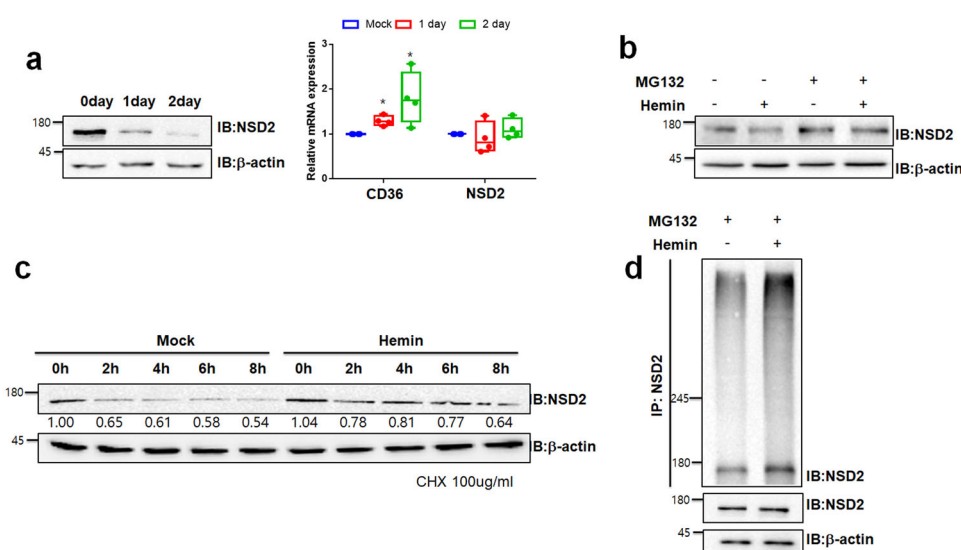

**Fig. 2 Differentiation of K562 reduces NSD2 stability through induction of ubiquitination. a** NSD2 protein level was confirmed by immunoblotting the lysates of K562 cells, treated with 30 μM hemin for 1 and 2 days (left). The levels of *CD36* and *NSD2* mRNA were analyzed using real-time PCR (right). Results are presented as mean ± SEM, $n = 4$; *$p < 0.05$. **b** NSD2 expression level was confirmed by immunoblotting using indicated antibodies. The lysates of K562 cells, were treated with 30 μM hemin for 2 days and MG132 proteasome inhibitor for 6 h. **c** Effects of hemin-mediated differentiation for regulation NSD2 stability were analyzed by immunoblotting using indicated antibodies. K562 cells were treated with 30 μM hemin for 16 h and treated with 100 μg/ml cycloheximide. Cell lysates were harvested after 2, 4, 6, and 8 h. **d** Immunoprecipitation analysis showed polyubiquitination chains of NSD2. K562 cells were treated with 30 μM hemin for 2 days and 10 μM MG132 for 6 h before harvested. Cell lysates were immunoprecipitated using anti-NSD2 antibody, and associated proteins and polyubiquitination chains were immunoblotted using indicated antibodies.

stability of NSD2 was reduced during NaBu-mediated differentiation (Supplementary Fig. 3). To further analyze the time-dependent *NSD2* expression in cells, we treated K562 cells with a protein synthesis inhibitor, cycloheximide (CHX; 100 μg/ml). Interestingly, NSD2 half-life in control cells was calculated to be about 8 h, and rapidly decreased to about 4 h in hemin-treated K562 cells (Fig. 2c). Further, hemin treatment resulted in increased polyubiquitination of NSD2 in K562 cells (Fig. 2d), suggesting that hemin-mediated K562 cell differentiation induces degradation of NSD2 as a proteasome-dependent manner.

**BRCA1 has E3 ligase activity toward NSD2.** From data obtained in a prior study, we identified NSD2 has been known to interact with an ubiquitin E3 ligase, BRCA1 (Supplementary Table 2)[28]. To validate the interaction between NSD2 and BRCA1, we performed co-immunoprecipitation (co-IP) assay in NSD2 and BRCA over-expressing 293T cells, which showed that NSD2 physically interacts with BRCA1 in vivo (Fig. 3a). Further, GST pull-down assay also confirmed the interaction. To investigate which domain of BRCA1

was involved in the interaction, we performed in vitro GST pull-down assay with BRCA1 wild type (WT) or deletion constructs (Fig. 3b). We identified that the middle region of BRCA1 (BRCA1 del#2) was responsible for the interaction between BRCA1 and NSD2 (Fig. 3c). Likewise, in vitro GST pull-down assay using NSD2 WT or deletion constructs suggested that the middle part of NSD2 (NSD2 del#2), which includes HMG, PHD, RING, and PWWP domains, contributes to the interaction between NSD2 and BRCA1 (Supplementary Fig. 4A, B). Since BRCA1 has E3 ubiquitin ligase activity, we next tested whether BRCA1 regulates NSD2 protein level. Stable BRCA1 knockdown 293T cells were created by lentiviral infection using two independent shRNAs. Knockdown of BRCA1 upregulated NSD2, while NSD2 was downregulated in BRCA1-overexpressed 293T cells (Fig. 3d, e). To investigate whether NSD2 stability is regulated via polyubiquitination during BRCA1 overexpression, we monitored time-dependent NSD2 protein level, with 100 μg/ml CHX, in BRCA1 overexpressing 293T cells. As expected, BRCA1 overexpression decreased stability of NSD2, but not that of control (Supplementary Fig. 4C). Fur-thermore, BRCA1 could enhance NSD2 polyubiquitination level in

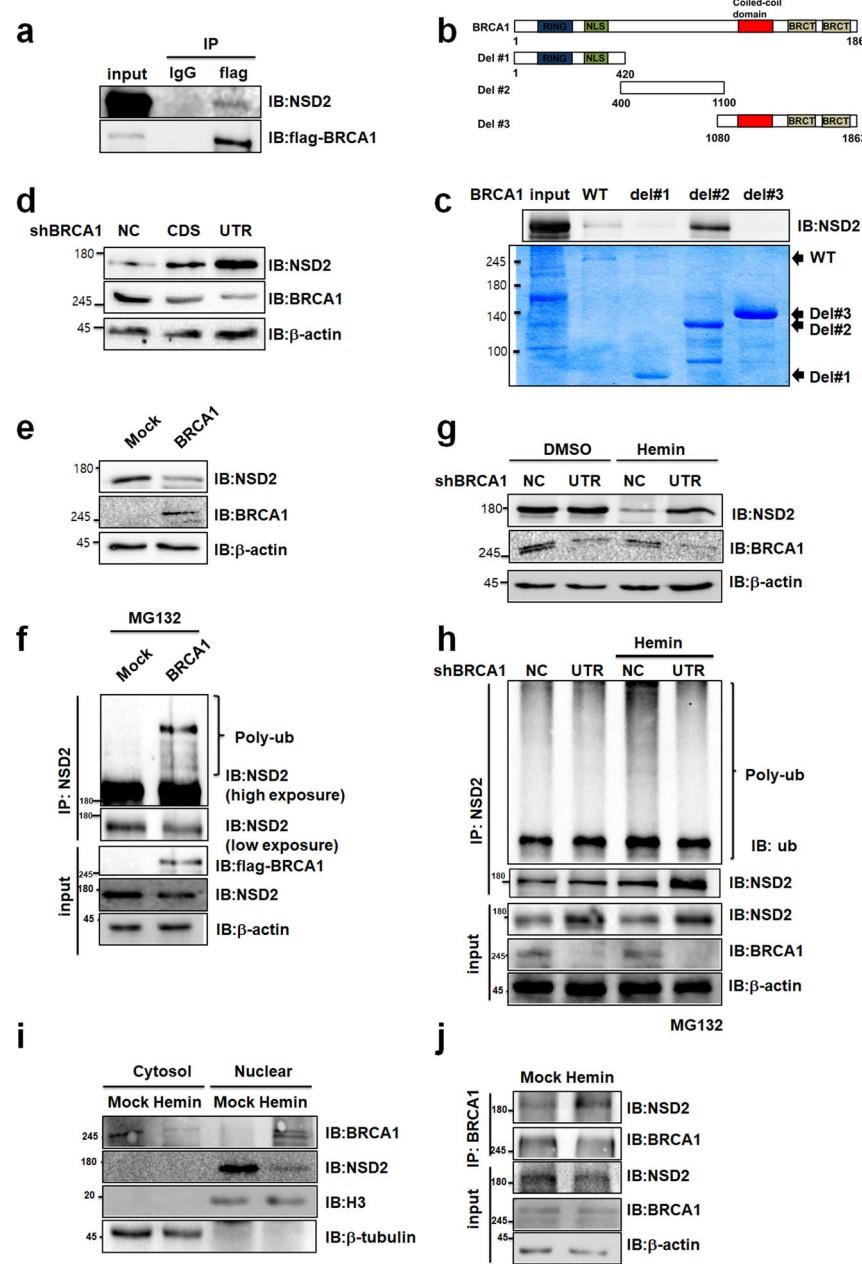

**Fig. 3 BRCA1 reduces NSD2 protein stability via induction of polyubiquitination of NSD2. a** BRCA1 interacting partners were immunoprecipitated with anti-flag antibody from flag-BRCA1 transfected 293T cell extracts. Associated proteins were eluted, resolved by SDS-PAGE, and immunoblotted. **b** Schematic representation of BRCA1 or BRCA1 deletion constructs. **c** Truncated GST-BRCA1 variants were used for GST pull-down. Extracts of K562 cells were incubated with GST-BRCA1 or GST-BRCA1 deletion mutants. Associated proteins were eluted, resolved by SDS-PAGE and immunoblotted (top panel). The amount of BRCA1 or BRCA1 deletion mutants were determined by Coomassie staining (Bottom panel). **d** Immunoblot analysis of NSD2 and BRCA1. 293T cells were stably BRCA1 knocked-down using two independent shRNAs. **e** Extracts from 293T cells transfected with BRCA1 were immunoblotted with indicated antibodies. **f** Immunoprecipitation analysis showed polyubiquitination chains of NSD2. 293T cells were transfected with BRCA1 and treated with 10 μM MG132 for 3 h before harvested. **g** NSD2 and BRCA1 protein levels were shown. K562 stable BRCA1 knockdown cells were treated with 30 μM hemin and immunoblotted with indicated antibodies. **h** Immunoprecipitation analysis showed polyubiquitination chains of NSD2. K562 BRCA1 stable knockdown cells were treated with 30 μM hemin and 10 μM MG132. **i** Localization of NSD2 and BRCA1 was measured using Immunoblot. Total proteins from K562 cells treated with 30 μM hemin for 2 days were separated into cytoplasmic and nuclear fractions. H3 and tubulin were used as positive controls for nuclear and cytoplasmic fractions, respectively. **j** NSD2 interacting partners were immunoprecipitated with anti-BRCA1 antibody from K562 cells treated with 30 μM hemin for 2 days and MG132 for 6 h.

293T cells (Fig. 3f). These results suggested that BRCA1 interacts with and promotes degradation of NSD2 via polyubiquitination. Next, we investigated whether BRCA1 regulates NSD2 in hemin-treated K562 cells. Consistently, erythrocyte differentiation by hemin decreased NSD2 protein level in control cells but not in

BRCA1 knockdown K562 cells (Fig. 3g). In addition, the poly-ubiquitination level of NSD2 increased during hemin-mediated differentiation in control K562 cells, but not in BRCA1 knockdown K562 cells (Fig. 3h). Surprisingly, we found knockdown of BRCA1 has no effect on endogenous NSD2 expression in undifferentiatied-

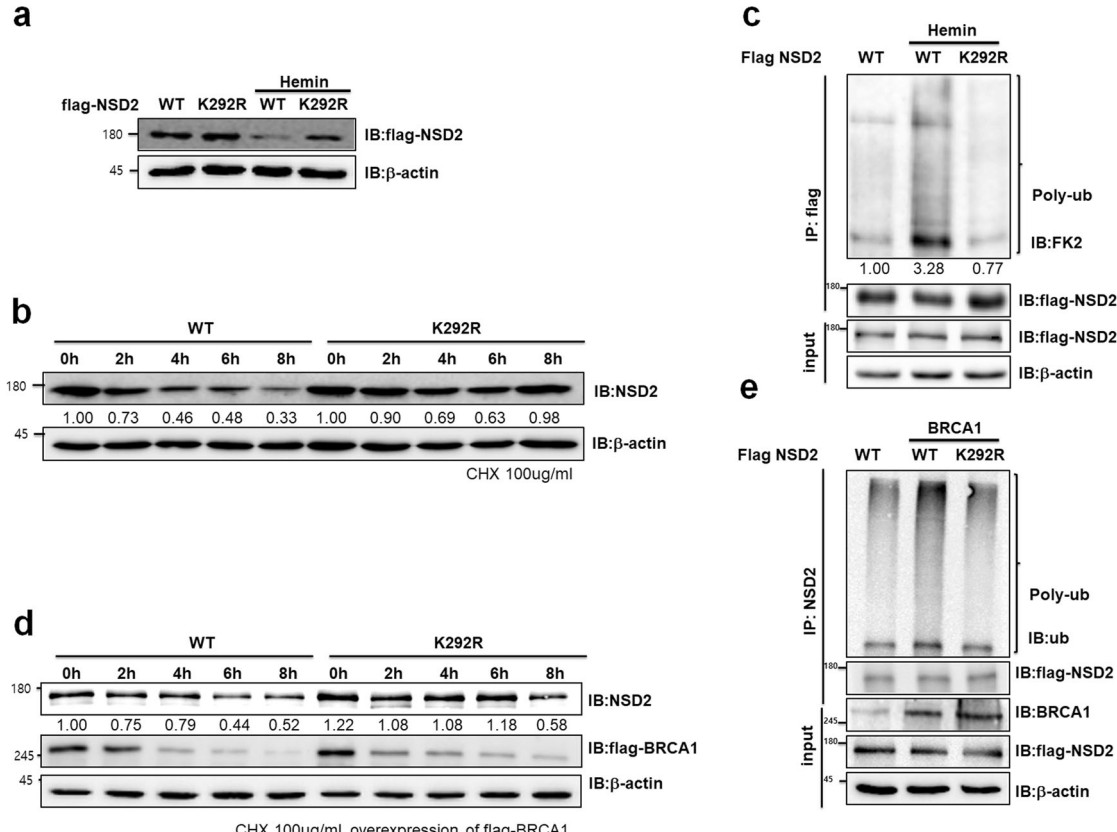

**Fig. 4 BRCA1 induces polyubiquitination of NSD2 during hemin-mediated differentiation. a** Immunoblot analysis of NSD2. K562 cells overexpressing either WT or K292R variant of NSD2 were treated with 30 μM hemin for 2 days. **b** K562 cells that stably overexpressed either WT or K292R variant of NSD2 were treated with 30 μM hemin and 100 μg/ml cycloheximide. Cell lysates were harvested after 2, 4, 6, and 8 h and analyzed by immunoblot using indicated antibodies. **c** Immunoprecipitation analysis showed polyubiquitination chains of NSD2. K562 cells overexpressing either WT or K292R variant of NSD2 were treated with 30 μM hemin for 2 days and 10 μM MG132 for 3 h. **d** 293T cells were transfected with BRCA1 and NSD2-WT or K292R variant, and treated with 100 μg/ml cycloheximide. Cell lysates were harvested after 2, 4, 6, and 8 h and analyzed by immunoblot using indicated antibodies. **e** Immunoprecipitation analysis showed polyubiquitination chains of NSD2. 293T cells were transfected with BRCA1 and NSD2-WT or K292R variant, and treated with 10 μM MG132 for 3 h.

K562 cells unlikely 293T cells (Fig. 3g, h). Since we assumed that BRCA1 might not be co-localized with NSD2 in undifferentiated-K562 cells, we investigated the localization of NSD2 and BRCA1 during hemin-mediated K562 differentiation. NSD2 principally localized to nucleus and differentiation by hemin downregulated NSD2 (Fig. 3i). Consistently, NaBu enhanced the translocation of BRCA1 from the cytosol into the nucleus making it co-localize with NSD2 (Fig. 3i and Supplementary Fig. 4D). Therefore, endogenous interaction between BRCA1 and NSD2 also increased during hemin-mediated K562 cell differentiation (Fig. 3j). Taken together, BRCA1 has E3 ubiquitin ligase activity for NSD2 and reduces NSD2 protein level via translocation in hemin-treated K562 cells.

**BRCA1 induces polyubiquitination of NSD2.** Previous studies suggested that oxidative stress, such as $H_2O_2$, induces erythroid differentiation of K562 cells[29,30] and DNA damage induces polyubiquitination of NSD2 at Lys292 in proteomic database[31]. We identified that NSD2 expression level was decreased in several DNA damaging agent treatments including UV, etoposide, $H_2O_2$, and doxorubicin (Supplementary Fig. 5). Since BRCA1 was related to reduction of NSD2 protein stability during K562 differentiation and oxidative stress induces K562 differentiation[30], we hypothesized that BRCA1 might induce polyubiquitination of

NSD2 at Lys292 during hemin-mediated differentiation. First, we generated flag-NSD2 or flag-NSD2 K292R overexpressing stable K562 cells and observed that NSD2 is downregulated, but not the ubiquitination-deficient mutant, NSD2 K292R, during hemin-mediated differentiation (Fig. 4a). To investigate whether NSD2 stability is regulated via polyubiquitination at Lys292 during hemin treatment, we monitored time-dependent NSD2 expression, with 100 μg/ml CHX, in hemin-treated NSD2 WT or NSD2 K292R overexpressing K562 cells. As expected, hemin-induced K562 differentiation decreased stability of NSD2 WT, but not that of NSD2 K292R (Fig. 4b). We next treated MG132 and identified the poly-ubiquitin chain on NSD2 and NSD2 K292R in hemin-treated K562 cells. Treatment of hemin induced polyubiquitination level of NSD2, but not in NSD2 K292R (Fig. 4c). To understand the role of BRCA1 in regulation of NSD2 polyubiquitination at Lys292, we estimated stability of WT or K292R mutant of NSD2 in BRCA1-overexpressed 293T cells after 100 μg/ml CHX treatment. Addition of BRCA1 destabilized NSD2 WT when 293T cells were treated with CHX, while NSD2 K292R mutant remained stable for up to 6 h (Fig. 4d). In addition, BRCA1 induced polyubiquitination of NSD2, but not that of NSD2 K292R (Fig. 4e). These results suggest that during K562 cells differentiation, BRCA1 regulates NSD2 stability via poly-ubiquitination of NSD2 at Lys292.

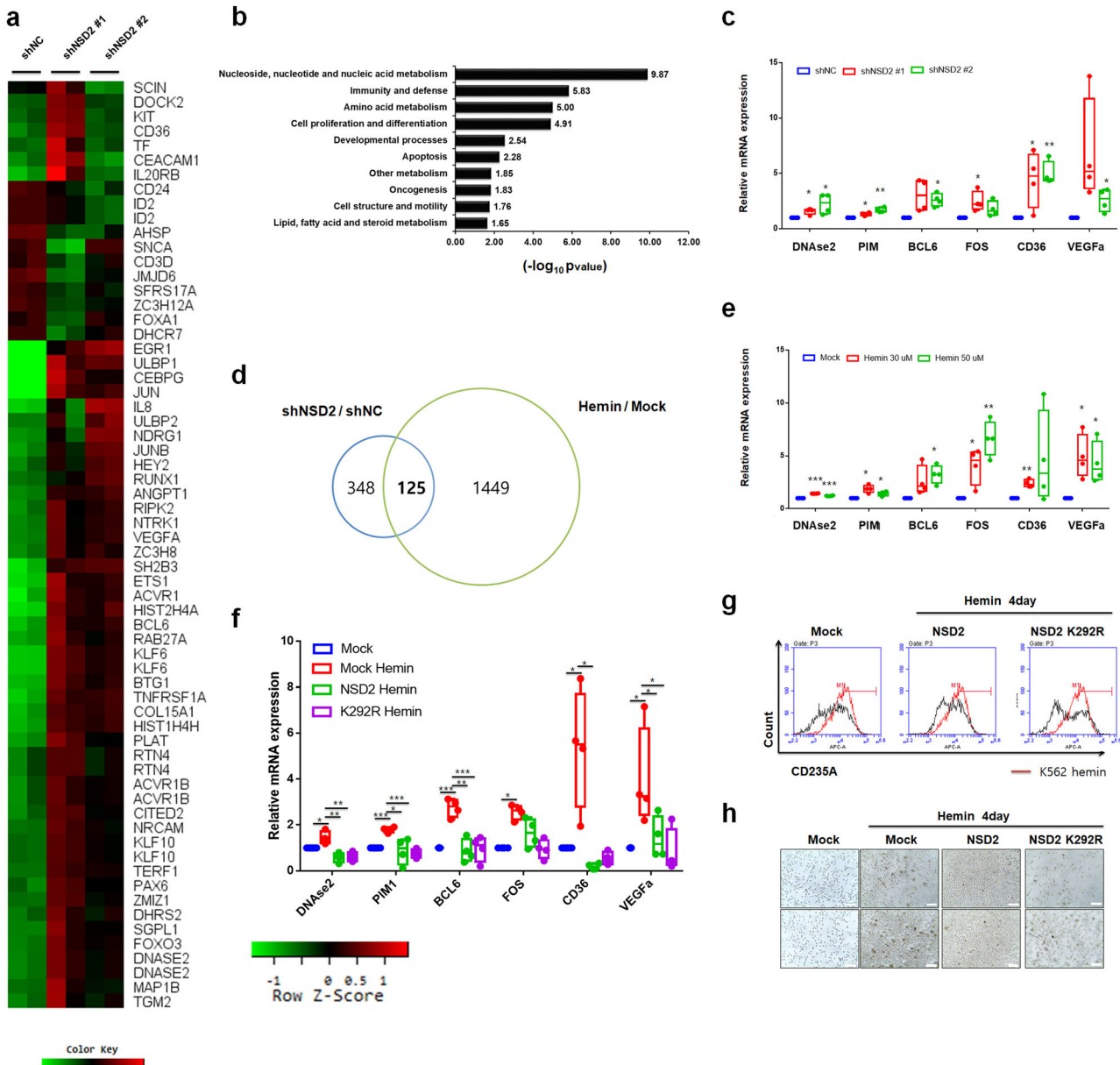

**Fig. 5 NSD2 disrupts K562 cell differentiation by regulating the expression of erythrocyte differentiation-related genes. a** Identification of NSD2-target genes by hierarchical clustering; differentially expressed hematopoietic-related genes in K562 NSD2-depleted cells are shown. Gene clusters, up- or downregulated by a factor of at least 1.5-fold, are represented in red and green, respectively. **b** Functional classification of NSD2-target genes by the annotation tool, DAVID. **c** The mRNA levels of differentiation-related genes in K562 stable NSD2 knockdown cells were quantified by real-time PCR and normalized to that of β-actin. Results are presented as mean ± SEM, $n = 4$; *$p < 0.05$, **$p < 0.01$. **d** Area-proportional Venn diagram showing the 125 overlapped in differentially expressed genes between NSD2 knockdown and hemin treatment in microarray data. **e** K562 cells were treated with 30 or 50 μM concentrations of hemin and the mRNA levels of differentiation-related genes were quantified by real-time PCR and normalized to those of β-actin. Results are presented as mean ± SEM, $n = 4$; *$p < 0.05$, **$p < 0.01$, ***$p < 0.001$. **f** K562 cells overexpressing either WT or K292R variant of NSD2 were treated with 30 μM hemin for 2 days. The mRNA levels of differentiation-related genes were quantified by real-time PCR and normalized to those of β-actin. Results are presented as mean ± SEM, $n = 4$; *$p < 0.05$, **$p < 0.01$, ***$p < 0.001$. **g** Cell differentiation by hemin was measured by staining K562 cells overexpressing WT or K292R variant of NSD2, or in vector control with CD235A for 1 h and sorted by FACS. The value of hemin-treated K562 cells (red line) was used as a control. **h** Cell differentiation was measured by staining K562 cells overexpressing WT or K292R variant of NSD2, or in vector control with o-dianisidine. Cells stained in brown (black arrows indicate o-dianisidine-positive cells) indicate hemoglobin accumulation. Scale bars, 25 μm. All the results in (**c**), (**d**), and (**f**) are shown as means ± S.D. ($n = 3$). *$p < 0.05$, **$p < 0.01$, and ***$p < 0.001$.

### NSD2 regulates erythrocyte differentiation-related genes.

To understand the function of NSD2 during K562 cell differentiation, we profiled gene expressions in NSD2 stable knockdown K562 cells. The knockdown of NSD2 regulated 473 genes (fold change [FC] > 1.5, $p < 0.01$) and among them, 143 genes were downregulated and 331 genes were upregulated (Fig. 5a). We analyzed the pathways undertaken by NSD2-target genes, using DAVID ontology tool, and found that enrichment of target genes is involved in the regulation of metabolism, immunity, cell apoptosis, and differentiation (Fig. 5b). Since NSD2 act as a repressor of K562 differentiation into erythrocytes, we selected several target genes related to erythrocyte differentiation. We

performed qRT-PCR of selected genes, including *VEGFa, DNAse2, FOS, BCL6, PIM*, and *CD36*. Although NSD2 has HMTase activity for methylation of transcriptional activation marker, H3K36, differentiation-related target genes were strongly upregulated in *NSD2*-depleted K562 cells (Fig. 5c). Assuming that NSD2 might indirectly regulate selected genes, we performed chromatin immunoprecipitation (ChIP) analysis using anti-NSD2 and anti-H3K36me2 antibodies (Supplementary Fig. 6A). While recruitment of NSD2 at *BCL6, CD36*, and *HBE1* promoters was unaffected during hemin-mediated differentiation, it increased at a direct target gene, *RRAS2* promoter and induced H3K36me2 during hemin-mediated differentiation (Supplementary Fig. 6A), indicating that NSD2 indirectly regulates these target genes during K562 cell differentiation. Since NSD2 induces methylation of AURKA and enhances its kinase activity[10] and AURKA suppresses leukemic cell differentiation[32], we hypothesized that NSD2 regulates these genes via inducing AURKA kinase activity. We confirmed that methylation level of AURKA decreased during hemin-mediated differentiation (Supplementary Fig. 6B). AURKA inhibitor, alisertib accelerated K562 differentiation upon hemin treatment through inducing erythroid differentiation-related genes (Supplementary Fig. 6C–E).

Having evidence that knockdown of NSD2 induced erythrocyte differentiation (Fig. 1), we compared microarray data of NSD2 knockdown with that from a published hemin-treatment study[33]. About 26% of genes (125/473) which had more than 1.5-fold change in *NSD2* knockdown cells overlapped with hemin-inducible target genes (Fig. 5d). As expected, selected target genes were detected in the microarray data. To validate the regulation on selected genes in hemin-treated K562 cells, we performed qRT-PCR analysis. Consistently, hemin upregulated *BCL6, VEGFa, FOS, PIM1, DNaseII*, and *CD36* genes (Fig. 5e). Further, we investigated whether NSD2 overexpression disrupted hemin-mediated differentiation of K562 cells. Overexpression of *NSD2* suppressed hemin-mediated induction of differentiation-related genes (Fig. 5f). Furthermore, overexpression of *NSD2* K292R strongly downregulated hemin-inducible genes (Fig. 5f). To understand these differences in phenotypic levels, we measured *CD235A* level using FACS in *NSD2* overexpressed K562 cells following treatment of hemin. *NSD2* overexpression disrupted hemin-mediated *CD235A* induction and *NSD2* K292R showed stronger effect than the wild type (Fig. 5g). A similar result was obtained in o-dianisidine staining (Fig. 5h). Taken together, NSD2 disrupts hemin-mediated erythrocyte differentiation by downregulating erythroid differentiation-related genes.

**The role of BRCA1 and NSD2 on K562 cell differentiation**. To investigate the physiological significance of BRCA1-regulated NSD2 stability during K562 cell differentiation, we quantified expression level of the target genes, which are known to be downregulated by NSD2 in *BRCA1* knockdown K562 cells after hemin treatment. Consistently, hemin treatment upregulated erythrocyte differentiation-related genes, but could not affect target gene expressions in *BRCA1* knockdown K562 cells (Fig. 6a). Further, FACS analysis indicated that, compared with control cells, hemin-mediated differentiation was not observed in *BRCA1* knockdown cells (Fig. 6b). Similarly, BRCA1 knockdown resulted in lower number of o-dianisidine-positive cells (Fig. 6c). These results suggested that knockdown of *BRCA1* disrupts hemin-mediated differentiation by downregulating erythrocyte-related genes. Since various mutations in *BRCA1* have been identified in several cancers, including breast, ovarian, and lymphoid cancers[34], we hypothesized that downregulation of or mutations in *BRCA1* would induce leukemia through dysregulation of NSD2 during myeloid cell differentiation. Therefore, we

investigated the levels of *BRCA1* in lymphoma. HPA examination indicated that *BRCA1* is downregulated in lymphoma (Supplementary Fig. 7A). Consistently, tissue array demonstrated *BRCA1* downregulation in granulocyte and large B cell lymphoma (Supplementary Fig. 7B). Moreover, using TCGA database (https://portal.gdc.cancer.gov/) (Supplementary Fig. 7C), we identified several cancer-related mutations in *BRCA1* and selected common mutations to investigate their role in the degradation of NSD2 during K562 cells differentiation. First, we generated BRCA1 WT and mutant constructs overexpressing stable K562 cells and observed NSD2 level during hemin-mediated K562 cells differentiation. Consistent with previous data, while BRCA1 WT reduced NSD2 protein level, K1183R mutant did not (Fig. 6d). To investigate why BRCA1 K1138R mutant did not reduce NSD2 protein level during differentiation, we tested the interaction between NSD2 and BRCA1 in 293T cells. NSD2 interacted BRCA1 independent of the BRCA1 mutations (Supplementary Fig. 7D). Surprisingly, BRCA1 K1138R did not translocated from cytosol into nucleus and did not reduce protein level of NSD2 during K562 cell differentiation (Fig. 6e). In addition, BRCA1 WT induced interaction with NSD2 in response to hemin treatment but the other 4 mutants of BRCA1 did not interact with NSD2, suggesting mutant forms of BRCA1 did not translocate for interaction (Fig. 6f). To further investigate whether mutations of BRCA1 influence K562 cell differentiation, we measured CD235A levels using FACS in BRCA1 WT or each mutant overexpressing K562 cells following treatment with hemin. *BRCA1 WT* overexpression strongly enhanced hemin-mediated *CD235A* induction and *BRCA1 P871L, E1038G*, or *S1613G* slightly enhanced hemin-mediated *CD235A* induction, on the other hand BRCA1 K1138R fail to show the induction (Fig. 6g). Consistent result was obtained in o-dianisidine staining (Supplementary Fig. 7E). Taken together, level of BRCA1 is important in K562 cell differentiation and BRCA1 mutation (K1183R) discovered in lymphoma patients prevented its translocation into the nucleus in hemin-treated K562 cells and eventually disrupted K562 cell differentiation.

## Discussion
Erythroleukemia cell line K562 maintains common progenitor stage of the hematopoietic stem cell differentiation. The erythroid differentiation of K562 could be achieved by exposure to several pharmacological agents such as hemin, butyric acid, and anthracycline antitumor drugs[33,34]. These drugs generate oxidative stress in early differentiation process and oxidative damage induces erythroid terminal differentiation[30,32]. To investigate the relationship between hemin and oxidative stress in K562 differentiation, we first carried out CRISRP-Cas9 screening during hemin-mediated erythroid differentiation. We identified NSD2 as a regulator of erythroid differentiation in K562 cells. NSD2 is emerging as critical target that controls maintenance of leukemic cell and neuroblastoma cell differentiation[13,35]. Human protein atlas database suggested the overexpression of NSD2 in bone marrow cancer and lymphoma. To gain insight into the role of NSD2 in human erythroid differentiation, we overexpressed NSD2 in the erythroleukemic cell K562. NSD2 inhibited hemin-mediated erythroid terminal maturation significantly. Our microarray data suggested that NSD2 downregulated transcription of erythroid differentiation-related genes which are overlapped with hemin-inducible genes. Knockdown of NSD2 accelerated differentiation in hemin-treated K562 cells. Reciprocally, K562 cell differentiation to erythrocyte decreased NSD2 protein level via induction of polyubiquitination of NSD2 at K292. It was similar that neuroblastoma cell differentiation decreased NSD2 level through proteosomal degradation[13]. Our previous proteomic data suggested that NSD2 might interact with

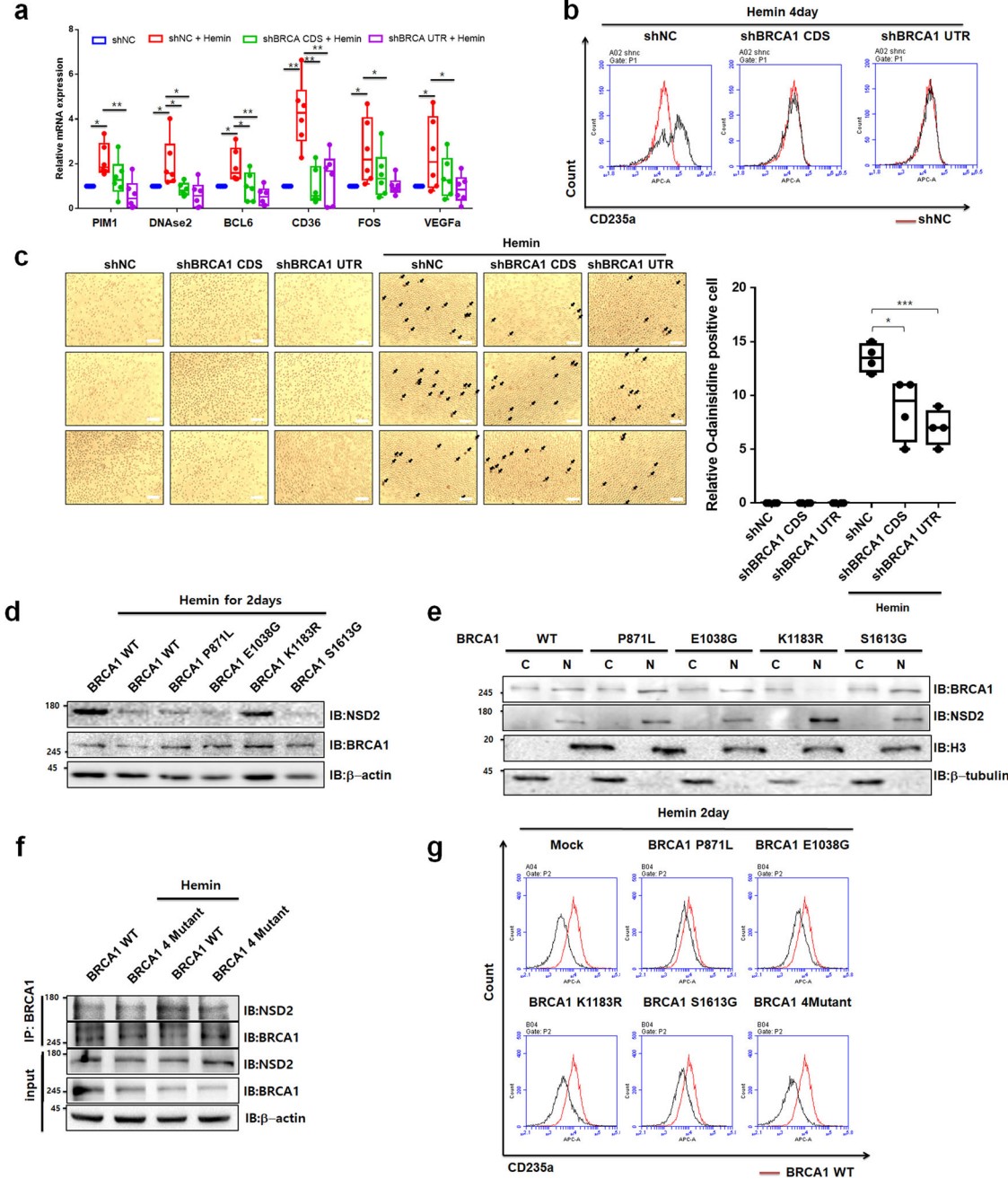

**Fig. 6 BRCA1 mutations disrupt hemin-mediated K562 cell differentiation. a** BRCA1 knockdown K562 cells were treated with 30 μM hemin for 2 days. The mRNA levels of differentiation-related genes were analyzed by real-time PCR and normalized to those of β-actin. The results are shown as means ± S.D. ($n = 6$). *$p < 0.05$, **$p < 0.01$, and ***$p < 0.001$. **b** Cell differentiation by hemin was measured by staining K562 BRCA1 knockdown or control cells, with CD235A for 1 h and sorted by FACS. The value of shNC K562 cells (red line) was used as a control. **c** Cell differentiation was measured by staining K562 BRCA1 knockdown or control cells, with o-dianisidine. Cells stained in brown (o-dianisidine-positive cells) indicate hemoglobin accumulation (left). Scale bars, 25 μm. Quantification data was shown (right). Results are presented as mean ± SEM, $n = 4$; *$p < 0.05$, ***$p < 0.001$. **d** Immunoblot analysis of BRCA1 and NSD2. K562 cells overexpressing either WT, P871L, E1038G, K1183R, or S1613G variant of BRCA1 were treated with 30 μM hemin for 2 days. **e** Localization of each mutant of BRCA1 was measured using Immunoblot. K562 cells overexpressing either WT, P871L, E1038G, K1183R, or S1613G variant of BRCA1 were treated with 30 μM hemin for 2 days. Total proteins were separated into cytoplasmic and nuclear fractions. H3 and tubulin were used as positive controls for nuclear and cytoplasmic fractions, respectively. **f** Interaction between NSD2 and BRCA1 WT or mutant was confirmed using immunoprecipitation analysis. K562 cells overexpressing either WT or 4 mutant variants of BRCA1 were treated with 30 μM hemin for 2 days and MG132 for 6 h. **g** FACS analysis showed K562 differentiation in WT, P871L, E1038G, K1183R, S1613G, or 4 mutants variant of BRCA1 overexpressing K562 cells. The K562 cells were treated with 30 μM hemin for 2 days. The value of hemin-treated K562 cells overexpressed BRCA1 WT (red line) was used as a control.

E3 ubiquitin ligase, BRCA1. We tested and identified the interaction between NSD2 and BRCA1 in this study. In addition, hemin induced the nuclear localization of BRCA1 and the increase of BRCA1-mediated NSD2 polyubiquitination. We also identified that

BRCA1 worked as an E3 ubiquitin ligase and increased polyubiquitination of NSD2 during K562 cell differentiation. Furthermore, knockdown of BRCA1 disrupted K562 cell differentiation via inhibiting NSD2 degradation. Non-functional mutations of BRCA1

which discovered in hematopoietic cancer also disturbed normal differentiation (Supplementary Fig. 7D). TCGA database revealed that NSD2 and BRCA1 had a number of common mutations in different types of cancers and they have been accumulated in cancer cell. Our study suggested that BRCA1 mutation disturbed its nuclear localization and interaction with NSD2 during erythroid differentiation (Supplementary Fig. 7F). Reciprocally, a mutation of NSD2 at K292 could inhibit its degradation through substitution of ubiquitination site. Therefore, our research provided a clue to how mutations of BRCA1 and NSD2 resulted in cancer transformation.

Previous studies show that NSD2 and BRCA1 played roles in DNA damage pathway such as NHEJ and HR[7,36,37]. NSD2 enhanced DNA damage repair through recruitment of 53BP1 to damage site. On the other hand, BRCA1 antagonized 53BP1 signaling to make HR pathway be dominant in repair process. BRCA1 depletion forced cells to rely on alternative repair pathway such as NHEJ. Here, we identified that NSD2 was degraded in several DNA damaging agent treatments including UV, Etoposide and $H_2O_2$ (Supplementary Fig. 5), and BRCA1 had an E3 ubiquitin ligase activity for NSD2 at K292 (Figs. 3 and 4). Therefore, we hypothesized that BRCA1 not only inhibited 53BP1 signaling but also could decrease NSD2 stability at DNA damage site for switching from NHEJ to HR. For identification of detailed mechanism for switching of DNA damage repair systems, further investigation is required. In this study, we found BRCA1 interact with NSD2 and reduces NSD2 protein stability. Besides DNA damage repair pathway, NSD2 functions as a master regulator of transcriptional activation, repression, and oncogenesis[38]. Especially, NSD2 stimulates myeloma cell growth though inducing proto oncogene, c-Myc expression[39]. Previous study indicated BRCA1 binds c-Myc and disrupts its transcriptional activity[40]. We confirmed BRCA1 and NSD2 involved not only in leukemia cell differentiation but also tumorigenesis via regulating c-Myc.

NSD2 is known for transcriptional activator by inducing methylation of histone H3 at Lys36. Surprisingly, our current study showed depletion of NSD2-induced erythroid differentiation-related gene expression. Previously, we discovered that NSD2 has a methyltransferase activity to non-histone protein AURKA and methylated AURKA enhanced its kinase activity[10]. Since methylation of AURKA decreased during K562 cell differentiation (Supplementary Fig. 6B), we hypothesized that hemin-mediated differentiation reduces the NSD2 stability, followed by decreasing AURKA activity. The inhibition of AURKA activity could affect K562 cell differentiation upon hemin treatment through activating erythroid differentiation-related genes (Supplementary Fig. 6). This process is quite similar to the fact that inhibition of AURKA induced PMA-mediated leukemic cell line, THP-1 differentiation[41].

Here, we uncovered innate differentiation-related factor, NSD2, and found that NSD2 regulated erythrocyte differentiation-related genes and disrupted hemin-mediated differentiation. We also identified that BRCA1 decreased NSD2 protein stability during K562 cell differentiation, and non-functional mutation of BRCA1 disrupted terminal differentiation of leukemic cells. Taken together, we suggested that regulation of NSD2 and BRCA1 as a candidate mechanism induced terminal differentiation for therapeutic possibilities.

## Methods

**Plasmid constructs**. The plasmid pCMV3-C-Flag NSD2 was purchased from Sino Biological Inc. (Beijing, China). GST- NSD2 fusion proteins and mammalian expression vectors were constructed by insertion of NSD2 into the bacterial expression vector pGEX-4T1, a pcmv-3Xflag vector (Sigma, St. Louis, MO), and modified plenti-puro-3Xflag vector (Addgene; #39481). The NSD2 (Y1118A) or (K292R) point mutant was constructed and inserted into the pcmv-3Xflag vector and modified plenti-puro-3Xflag vector. Sequences encoding NSD2 #1 (residues 1–400), NSD2 #2 (residues 400–1365), and NSD2 #3 (residues 1000–1365) were subcloned into pGEX-4T1 vectors. The plasmid pBABE puro HA BRCA1 was

purchased from Addgene (#14999). BRCA1 was also inserted into the bacterial expression vector pGEX-4T1 and plenti-puro vector. BRCA1 point mutants (P871L, E1038G, K1183G, S1613G, and 4 mutants) were constructed and inserted into the plenti-puro vector. Sequences encoding BRCA del#1 (residues 1–420), BRCA1 del#2 (residues 400–1100), and BRCA1 del#3 (residues 1080–1863) were subcloned into the and pGEX-4T1 vectors. The plasmids plko-shNSD2 #1, plko-shNSD2 #2, and pcDNA6-HA-Ub have been described previously[28]. The shRNAs against human BRCA1 were designed using siRNA sequence designer software (Clontech). A double-stranded oligonucleotide for shRNA plasmid construction was produced using primers from the 5′ to the 3′ end (Supplementary Table 1). These oligonucleotides were inserted into the AgeI/EcoRI site of the pLKO.1 TRC vector.

**Antibodies**. Antibodies against H3K36me2 (Millipore, Billerica, MA; 07-274, dilution (1:2000)), BRCA1 (07-434 dilution (1:1000)), Flag (Sigma, St. Louis, MO; F3165, dilution into 1 ng/ml)), NSD2 (EpiCypher, Durham, NC; 13-0002 dilution (1:1000)), MMSET/NDS2 (Abcam, ab75359 dilution (1:1000)), APC-CD235A (eBioscience, Waltham, MA;17-9987-42), FK2 (Enzo Life Sciences, Farmingdale, NY; BML-pw-8810-0100, dilution (1:1000)), β-actin (sc-47778 dilution (1:1000)), H3 (sc-8654, dilution (1:1000)), PARP1 (sc-56197, dilution (1:1000)), and tubulin (sc-9103, dilution (1:1000)) (all from Santa Cruz Biotechnology, Dallas, TX) were employed.

**Cell culture**. 293T was grown in Dulbecco's modified Eagle's medium (DMEM), and K562 in RPMI 1640, each containing 10% heat-inactivated fetal bovine serum and 0.05% penicillin-streptomycin, at 37 °C in a 5% $CO_2$ atmosphere. 293T cells were transfected with indicated DNA constructs using polyethyleneimine (PEI) or Lipofectamine 2000 (Invitrogen, Carlsbad, CA). For differentiation of K562 cells were treated with 30 mM hemin (Sigma, St. Louis, MO). After incubation for 48 h, cells were harvested and used in experiments. For confirming protein stability, the cells were treated with 100 μg/ml cyclohexamide (Sigma, St. Louis, MO). After incubation for 2, 4, 6, and 8 h, the cells were harvested and used in western blot assay.

**CRISPR-Cas9 screening**. CRISPR-Cas9 screening was performed as described in the literature[22]. For a schematic illustration of the experimental design for the genome-wide CRISPR screening, see Fig. 1a. The human Brunello pooled library plasmids (total sgRNA 76,441 and 4 sgRNA per a gene) and CRISPR-V2 plasmids (containing Cas9) were obtained from Addgene (cat #73178 and #52961, respectively) and amplified according to the recommended protocols by electroporation. To produce lentivirus, 293T cells were transfected in tissue culture dishes at a density equivalent to $2 \times 10^6$ cells per 100 mm surface area. Plasmid DNA was diluted into medium with a lentiviral packaging plasmid mixture of pMD2.G and pPAX2 and transfected using polyethylenimine (PEI) (3 μg PEI:1 μg DNA). Viral supernatants were collected between 36 and 72 h time points after transfection. Lentiviral titer was determined by transducing the cell line of interest plated at clonogenic density with serial dilutions of virus in the presence of 1 μg/mL puromycin. CRISPR-Cas9 screening was performed with 2-vector system. First, K562 cells were infected with lenti-CRISPR-V2 virus and selected using 1 μg/mL puromycin for making Cas9-expressing K562 cells. Cas9-expressing K562 cells were infected with Brunello pooled library virus (at a multiplicity of infection of 0.3) using polybrene at a density equivalent to $2.5 \times 10^7$ cells (~300 cells per a sgRNA) per 150-mm plate area. After 3 days, we cultured the K562 cells of $2.5 \times 10^7$ per each 150-mm plate and treated with NaOH or hemin for 3 days. For differentiated cell sorting, hemin-treated cells were washed with PBS, resuspended cells in 1X binding buffer, stained with CD235A-APC Abs for 30 min at RT in the dark.

The CD235A+ cells were separated on FACS Aria III (BD biosciences) and positive APC staining (total CD235A+ cells). Purity of sorted fractions as verified by FACS analysis was more than 50%.

**NGS library preparation and sequencing**. The sequencing library is prepared by PCR amplification methods. Genomic DNA (gDNA) was isolated using wizard genomic DNA purification kit (Promega) according to the manufacturer's protocol. For NGS library sequencing, we set up 10 parallel PCRs for a given sample. PCR of gDNA was performed to attach sequencing adaptors and barcode samples which were divided into multiple 50 μl reactions (total volume) containing a maximum of 5 μg gDNA, P5 stagger primer mix (1 μM), a uniquely barcoded P7 primer (1 μM), and 5X HiPi PCR master mix (Elpis Biotech). PCR cycling conditions: an initial 1 min at 95 °C; followed by 30 s at 94 °C, 30 s at 52.5 °C, 30 s at 72 °C, for 28 cycles; and a final 10 min extension at 72 °C. P5/P7 primers (Supplementary Table 1) were synthesized at Integrated DNA Technologies (IDT). Samples were purified with Agencourt AMPure XP SPRI beads according to manufacturer's instructions (Beckman Coulter, A63880). The final purified product is then quantified using DNA Picogreen according to the DNA Picogreen Quantification Protocol Guide (QuantiFluor dsDNA system kits for Promega) and qualified using the TapeStation DNA screentape HSD1000 (Agilent). And then we sequenced using the HiSeq™ 2500 platform (Illumina, San Diego, USA). Reads were counted by first searching for the CACCG sequence in the primary read file that appears in the vector 5′ to all sgRNA inserts. The next 20 nts are the sgRNA insert, which was then mapped to a reference file of all possible sgRNAs present in the library. The read was then

assigned to a condition (e.g., a well on the PCR plate) on the basis of the 8-nt barcode included in the P7 primer. The resulting matrix of read counts was first normalized to reads per million within each condition by the following formula: reads per sgRNA/total reads per condition $\times 10^6$. Reads per million was then $\log_2$-transformed by first adding 1 to all values, which is necessary in order to take the log of sgRNAs with zero reads.

**Data processing and analysis**. The sgRNA spacer sequences were than mapped to the reference human Brunello library. Normalized read counts were obtained by normalizing to total read count per sample. We first selected genes that more than three out of four sgRNAs showed common incremental or decreamental patterns compared with CD235A-positive cells. CRISPR guide scores were generated by calculating the averaging log fold change of normalized sgRNAs lead counts between NaOH or hemin-treated CD235A-positive samples.

**Immunoprecipitation and ubiquitination assays**. 293T cells were transfected with flag-BRCA1. After 48 h, cell lysates were immunoprecipitated using an NSD2 antibody. Protein A/G agarose beads (GenDEPOT) were then added, and the mixture was rotated for 2 h at 4 ℃. Bound proteins were analyzed by immunoblotting with indicated antibodies. For ubiquitination assays, transiently transfected 293T and K562 cells were lysed in modified RIPA buffer (10 mM Tris-HCl [pH 7.5], 150 mM NaCl, 0.025% SDS, 1% sodium deoxycholate, 1% NP-40, 1× protease inhibitor cocktail, 5 mM EDTA). The cell lysates were immunoprecipitated using anti-NSD2. Protein A/G agarose beads were then added, and the mixture was rotated for 2 h at 4 ℃. Bound proteins were analyzed by immunoblotting using the indicated antibodies.

**Reverse transcription and real-time PCR**. Total RNA was isolated from viral infected K562 cells using RNAiso Plus (TaKaRa, Kusatsu, Japan). After cDNA synthesis, cDNA was quantified and subjected to analysis of mRNA expression. The PCR primers used are presented in Supplementary Table 1. Dissociation curves were examined after each PCR run to ensure amplification of a single product of the appropriate length. The mean threshold cycle ($C_T$) and standard error values were calculated from individual $C_T$ values obtained from triplicate reactions per stage. The normalized mean $C_T$ value was estimated as $\Delta C_T$ by subtracting the mean $C_T$ of β-actin. $\Delta\Delta C_T$ values were calculated as the difference between the control $\Delta C_T$ and the values obtained for each sample. The $n$-fold change in gene expression, relative to an untreated control, was calculated as $2^{-\Delta\Delta CT}$.

**Subcellular fractionation**. Preparation of nuclear and cytosolic fractions was carried out by lysing cells for 10 min on ice using buffer A (10 mM HEPES [pH 7.9], 10 mM KCl, 0.1 mM EDTA, 1 mM DTT, 0.5 mM PMSF, 1× protease inhibitor cocktail, and 0.4% NP-40), followed by centrifugation at $15,000 \times g$ for 3 min. Supernatants were retained as cytosolic fractions, whereas the pellets were subjected to further lysis in buffer B (20 mM HEPES [pH 7.9], 0.4 M NaCl, 1 mM EDTA, 10% glycerol, 1 mM DTT, 0. PMSF and 1× protease inhibitor cocktail). The pelleted material was then resuspended by pipetting. After a 2 h agitation at 4 ℃, lysates were centrifuged at $15,000 \times g$, and the resulting supernatants were collected as nuclear fractions.

**FACS analysis**. Flow cytometry analysis to measure cell proliferation were performed as described previously[42] with some modification. pLKO-GFP-NC or pLKO-GFP-shNSD2 plasmids were used to generate lentivirus by transfecting 293T cells with pMD2.G and pPAX2 plasmids using PEI reagent. Viral supernatants were collected between 36 and 72 h time points following transfection. For proliferation assays, K562 were subjected to GFP lentiviral transduction and flow cytometry analysis using a BD Accuri™ C6 Plus flow cytometer (BD Biosciences). Gating was performed on live cells using forward and side scatter, prior to measuring of GFP positivity.

To measure the effect of NSD2 on apoptosis, K562 shNC and shNSD2 cells were washed with PBS, resuspend cells in 1X binding buffer, added FITC-Anenexin V and PI (BD Bioscience) for 30 min at RT in the dark. The cells were subjected to FACS analysis using a FACSCalibur system.

To measure the ratio of K562 differentiation, K562 cells were induced by NSD2 WT, Y1118A, or K292R, and treated with 30 mM hemin. The cells were washed with PBS, resuspend cells in 1X binding buffer, added APC-CD235A for 30 min at RT in the dark. The cells were subjected to FACS analysis using a FACSCalibur system.

**Chromatin immunoprecipitation analysis**. ChIP analysis was performed as described in the literature[9]. Briefly, K562 cells were treated with 30 mM hemin for 48 h. The cells were cross-linked by addition of 1% formaldehyde to the medium and incubation for 10 min at 37 ℃, followed by addition of 125 mM glycine and incubation for 5 min at room temperature. The cells were then lysed in SDS lysis buffer, and the samples were sonicated and immunoprecipitated using the indicated antibodies. The immunoprecipitates were eluted and reverse cross-linked. The DNA fragments were then purified and PCR amplified for quantification using each PCR primer pair (Supplementary Table 1). Disassociation curves were

generated after each PCR run to ensure amplification of a single product of the appropriate length. The mean threshold cycle ($C_T$) and standard error values were calculated from individual $C_T$ values, obtained from duplicate reactions per stage. The normalized mean $C_T$ value was estimated as $\Delta C_T$ by subtracting the mean $C_T$ of the input from that of each gene.

**O-dianisidine staining**. Hemoglobin activity was detected in K562 cells by performing o-dianisidine staining as described[43]. Briefly, K562 cells were cultured with 30 nM hemin for 2 days. To detected hemoglobin synthesis, K562 cells were washed with PBS and stained for 15 min in the dark in 0.6 mg/ml 3,3′-dimethoxy benzidine (Sigma, St. Louis, MO, USA), 0.01 M sodium acetate (pH 4.5), 0.65% $H_2O_2$ and 40% (v/v) ethanol. The cells were then washed once with PBC and suspended with 70% glycerol in glass slides for microscopy. The percentage of dianisidine-positive cells were calculated by counting three independent fields to obtain the average. For some experiments, K562 cells were co-treated with alisertib (AURKA inhibitor).

**Tissue array**. Formalin-fixed, paraffin-embedded tissue array slides containing lymphoma tissues (BIOMAX; SP482) was purchased from US BIOMAX. Briefly, after deparaffinization in xylene and rehydration in graded ethanol, endogenous peroxidase activity was blocked by incubating with 3% hydrogen peroxide for 10 min. Next, tissue sections were heated in 100 mM citrate buffer (pH 6.0) for 10 min to retrieve antigens and then preincubated with normal horse serum for 20 min at room temperature. Anti-BRCA1 antibodies (diluted 1:100) were used as the primary antibodies. The specimens were subsequently incubated with biotinylated anti-rabbit and anti-mouse secondary antibody (Vectastain Laboratories) and streptavidin–horseradish peroxidase (Zymed Laboratories Inc.). DAB (3,3-diaminobenzidine; Vectastain Laboratories) was used as a chromogen, and eosin was used for counterstaining.

**Microarray analysis**. For *NSD2*-target gene profiling, we used the Illumina HumanHT-12 v4 Expression BeadChip (Illumina), which includes a bead pool of more than 47,231 unique bead types corresponding to 28,688 RefSeq annotated transcripts. Total RNA (0.55 μg) isolated from control and shNSD2 stable K562 lines was reverse transcribed and amplified according to the protocols described in the Illumina TotalPrep RNA amplification kit manual (Ambion). In vitro transcription was then carried out to generate cRNA (0.75 μg), which was hybridized onto each array (two replicates for each condition) and then labeled with Amersham fluorolink streptavidin-Cy3 (GE Healthcare Bio-Sciences). The array was then scanned using the Illumina Bead Array Reader Confocal Scanner. Array data export processing and analysis were performed using Illumina GenomeStudio v2011.1 (Gene Expression Module v1.9.0). This data set was submitted to the Gene Expression Omnibus under submission number GSE144939. Array probes were transformed by logarithm and normalized by quantile method. Gene enrichment and functional annotation analysis for the significant probe list were performed using the DAVID software (http://david.abcc.ncifcrf.gov/home.jsp).

**Statistics and reproducibility**. Data are expressed as mean ± SEM of three or more independent experiments of mRNA expression, luciferase assay, and proliferation assay and ± SD of three technical triplicates from a representative experiment of ChIP assay. Statistical significance ($p < 0.05$) was calculated using Microsoft Excel. Differences between groups were evaluated by one-way analysis of variance (ANOVA), followed by a Student's $t$-test or Bonferroni test, as appropriate. We repeated at least four times experiments and the exact sample size ($n$) for each experiment appear in figure legend. Using the Boxplot with whiskers from minimum to maximum, we showed data distribution clearly and each points mean individual data points.

**Reporting summary**. Further information on research design is available in the Nature Research Reporting Summary linked to this article.

## Data availability

Microarray data have been deposited in NCBI with accession number GSE92878 and GSE133939. Raw data for graphs can be found in Supplementary Data 1. CRISPR sgRNA sequences are available in Supplementary Data 2. All other data are available within the manuscript files or from the corresponding author upon reasonable request.

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

## Acknowledgements

This work was supported by the Ministry of Education, Science and Technology (NRF-2019R1C1C1002231, NRF-2019R1A4A2001609, and NRF-2017R1A2B4004407) the Basic Science Research program through the National Research Foundation of Korea (NRF), Republic of Korea.

## Author contributions

S.B.S. and J.W.P. designed the experiments and analyzed the results. J.W.P., J.Y.K., J.Y.H., and H.J.K. performed experiments and analyzed the results. S.B.S. and J.W.P. wrote the paper.

## Competing interests

The authors declare no competing interests.
