## [Peer Review File · Communications Biology]

Reviewers' comments:

Reviewer #1 (Remarks to the Author):

In this manuscript, the study revealed a role for the histone methyltransferase MMSET in erythroid differentiation and uncovered a novel interaction between the tumor suppressor BRCA1 and MMSET. The authors demonstrated that ubiquitination of MMSET by BRCA1 and its subsequent degradation is required for heme-induced erythroid differentiation in K562 cells. The findings of the manuscript are of clinical relevance, taking into consideration the well-known role of MMSET in pathogenesis and chemoresistance in different hematologic malignancies. However, the evidence on the role of each of BRCA1 and MMSET in erythroid differentiation provided in the manuscript is not sufficient

Major Revisions:

1- In the performed genome-wide CRISPR screen, the authors only sorted cells that were treated with hemin and compared the sorted pool (CD235A+ cells) to the unsorted vehicle-treated pool. Using this strategy, it is not feasible to distinguish between genes whose loss by itself is sufficient to drive erythroid differentiation and those specifically involved in hemin-induced erythroid differentiation. The former set of genes can be identified by sorting vehicle-treated cells and identifying sgRNAs that are enriched in this population (the disruption of certain genes could result in CD235A+ cells even in the absence of hemin).

2- Why are cells treated with hydroxide- what kind of control is this- what concentration of NaOH? Line 101- what does "innate differentiation-associated factor mean?"

3- What do you mean by screening in the pan-cancer atlas of leukemia on line 102- reference 27 is a database of post-translational modifications of proteins and has nothing to do with leukemia.

4- Figure 1C is uninterpretable what does multiple splenic cancers mean. What disease is being studied in those sections. Similarly what tissues does figure S1C refer to. S1D shows high level expression of MMSET in lymphoma specimens- why is this data important and how does it support a role of MMSET in lymphoma or leukemia. The topic of this manuscript is erythroid differentiation.

5- In Results, line 110, the authors stated that "hemin-mediated differentiation decreased in MMSET-overexpressed K562 cells". However, the figures (Figure 1 D-F) supporting this statement are not very convincing. In Figure 1D, Hemin treatment seems to induce CD36 expression levels in MMSET overexpressing cells, but these levels are downregulated by MMSET overexpression even in non-treated controls. In figure 1E, the images have different background contrasts and the differences between o-dianisidine positive cells are not clear. Similarly, in figure 1F, the % of CD235A+ cells are not shown, and the differences are not clear. Quantification of the flow data from 3 independent experiments with statistics is necessary in this case. Alternatively, immunoblot for CD235A would clearly show the differences. The authors can also use other markers of erythroid differentiation such as globin genes (HBA and HBG) to support their hypothesis. The manuscript could also benefit from using other erythroid differentiation inducers such as sodium butyrate for these experiments. In the supplemental figure 2e shRNA knockdown of MMSET appeared to lead to differentiation even without hemin and accentuated hemin differentiation- this effect needs to be quantified and these data are stronger than the enforced expression data.

6- Figure 2- how many times was the cycloheximide chase repeated- can an estimate of the half-life of MMSET in the presence or absence of hemin be generated. Does other differentiation inducers cause this degradation as well? MMSET has a long and short isoform- only one form is shown- no MW markers are given- is the short form seen in 293 or K562 cells?

7- Line 149- reference 29 has no mention of BRCA1

8- The manuscript demonstrated evidence on the physical interaction between BRCA1 and MMSE; however endogenous co-precipitation was not performed and so whether the complex exists in cells is not yet shown. No complex is shown in K562 cells. The overexpression co-IP and GST pull

downs are supportive but not definitive. The overexpression of BRCA1 decreased MMSET levels in 293T cells but this has to be shown in K562 cells; the knockdown of BRCA1 in 293 cells increased MMSET levels but effect in K562 are shown. Supp figure 3C- Basal levels of MMSET are lower in BRCA1 overexpressing cells but the change in half life has to be calculated by quantifying several blots. Figure 3I overexpression of BRCA1 in 293 cells increased MMSET ubiquitylation- did this also happen in K562 cells. Again the results only from knockdown of BRCA1 in K562 cells is shown. Line 174 is very confusing after showing that shBRCA1 prevents hemin induced decrease of MMSET levels in figure 3g- the sentence says BRCA1 has no effect on endogenous MMSET levels. Was 3g then with overexpressed proteins or do you mean that BRCA1 knockdown does not affect MMSET levels before hemin addition- figure 3g then would need to be repeated a few times to convincingly show this particularly since there was some increase in benzadine staining in K562 cells with shRNA BRCA1 even before hemin. Supp figure 3d not very convincing co-localization since the MMSET levels are so much lower with hemin- there seems to be pretty good co-localization even before hemin. Hemin is stated to lead to transport of BRCA1 from cytoplasm to nucleus. The IF in supp 3d shows fialry prominent BRCA1 expression in nucleus and cytoplasm while the fractionation in 3i shows almost no nuclear BRCA1- there is no cytoplasmic loading control and was this done by loading equal fractions on the gel or equal ug of protein? 3J is with overexpressed MMSET and the co-IP with BRCA1 is not convincing- the experiment should be performed with endogenous co-IP. Line 181 "endogenous interaction" is no accurate.

9- Line 188- reference 32 does not mention MMSET unicityltion but there is a database resulting from this paper that indicates the polyubiquityltion. That being said figure 4 has some of the strongest data in the paper indicating that the K292 residue has a key role in stability of MMSET

10- In Figure 5, the authors selected a set of genes related to erythroid differentiation to confirm their induction by MMSET knockdown. However, key genes in erythroid differentiation including master regulators (GATA1/2), the transferrin receptor (TFRC) as well as globin genes were not addressed. Studying the regulation of such genes by MMSET would strongly support its role in erythroid differentiation. To show MMSET is specifically recruited to promoters- control PCR for regions 5', 3' to the promoter and control gene desert regions need to be performed.

11- In Figure 5G, a control where cells with the mock vector treated with hemin needs to be added.

12- Figure supp 5a, b as before what this protein atlas data refers to is unclear as is the term splenic cancers. Supp 5e- change "upnormal" to abnormal .

13- 6H what is "mock". None of the BRCA1 mutants accelerated (not the correct term; maybe enhanced) CD235a expression.

14- The authors need to discuss the revealed interaction between BRCA1 and MMSET in the context of their roles as a tumor suppressor and an oncogene respectively.

Minor Revisions:

1- Line 53- in vivo MMSET only modulates H3K36 dimethylation; other sites are found in vitro but not in vivo

2- Line 58- which studies showed a role for MMSET in leukemia cell differentiation- none are quoted by the authors

3- In figure 6F, MMSET and BRCA1 immunoblot panels seem to be reversed.

2- In line 225, it is not clear why the authors chose RRAS2 promoter for MMSET and H3K36me3 ChIP-PCR analysis. MMSET creates the H3K36me2 mark- why isn't this being tested?

3- In the "CRISPR-Cas9 Screening" methods section, line 393, the correct name of the used CRISPR library should be used. It should be "Brunello" instead of CRISPR-v2. The authors also

need to indicate that they used the lentiGuide-Puro library vector (2-vector system).

4- In the "NGS library preparation and sequencing" methods section, the authors need to indicate what was the criteria used for hit selection or what method was used for gene ranking.

5- In the "O-dianisidine methods section", the authors described an O-dianisidine protocol that is used for Zebrafish embryos, although they did not use Zebrafish in their study! I'm not sure what do they mean by "K562 cells were dechorionated"! The correct staining protocol used in this study should be added.

6- The manuscript has many grammar mistakes and many sentences that need to be rephrased to enhance the overall clarity.

Reviewer #2 (Remarks to the Author):

Jin Woo Park and colleagues in the Sang-Beom Seo laboratory performed a CRISPR screen looking for genes affecting the hemin-mediated differentiation of erythroleukemia cell line K562. They identified many genes but noted with interest the MMSET (WHSC1/NSD2) gene that the authors have a previous interest in. They also discovered MMSET in databases of other investigators' data and discovered possible interaction with BRCA1, and in another that MMSET is ubiquitinated on lysine 292 after DNA damage. These data are significant. Loss of BRCA1 function in murine hematopoietic cells leads to bone marrow failure/lack of hematopoietic cell differentiation, but the mechanism of that to my knowledge is unclear. Here, the authors find that MMSET is the target of ubiquitin mediated degradation by BRCA1 during K562 differentiation, and that over-expression of ubiquitin-resistant mutant of MMSET interferes with differentiation. Similarly, knock-down of BRCA1 has the same effect. These data add to our knowledge of MMSET biochemistry and its role in hematopoietic cell differentiation.

These are good novel data that add significantly to our understanding of MMSET biochemistry, but the manuscript is weakened by some clumsy wording and overly broad conclusions regarding the functional implications. An extended amount of text regarding MMSET's role in myeloma is not well-justified given that myeloma is not studied here. Also, with the focus of this manuscript on MMSET's role in maintaining an undifferentiated state in a myeloid cell line, it is not clear how that is relevant to myeloma only. Also, while the data appear convincing, statistical analyses are missing throughout.

Detailed critique

Abstract: Line 18, "Multiple myeloma SET domain (MMSET) is linked to the IgH promoter by t(4;14)(p14;q32)..." It's not clear to me why the authors focus on myeloma here given the upregulation of MMSET in other cancers too. The gene was first identified in a region deleted in a congenital syndrome as WHSC1, as the authors I'm sure are aware. The HGNC name is NSD2, which may reduce the confusion of readers. Line 26, " Proteomic data analysis revealed interaction.." the source for this analysis is not well presented. If this was another laboratory's database, this should be made clear. Line 35, "..as a potential therapeutic target process in myeloma." Again, why only myeloma if MMSET is upregulated in multiple cancers?

After presenting the results of their CRISPR screen, the authors begin the manuscript by providing data on MMSET from published cancer databases which is poorly presented and made worse by reaching for conclusions from those surveys that are not supported by the data. We know that MMSET is upregulated in a variety of cancers. Referring later to bone marrow cancers and lymphomas (very broad groups of diseases) is not helpful.

Line 101: "..innate differentiation-related factor, MMSET as a common candidate in both this

screening and TCGA pan-cancer atlas database of leukemia (Krassowski M 2018). Can the authors clarify? This reference does not appear to support this claim. Krassowski paper is on a different database ActiveDriverDB. Searching this database for WHSC1 reveals a smattering of somatic mutations of unclear significance. I also attempted to find MMSET in TCGA data sets and could not find support for the idea of MMSET as a "differentiation-related factor." Extending the findings that MMSET is a differentiation-related factor is a strength of this work.

Line 105: "..MMSET is upregulated during lymphoma." This is poor wording. MMSET is over-expressed in lymphoma samples compared to what? Normal lymphoid tissue?

Line 110: "..suggesting that overexpression of MMSET is related to leukemia maintenance." Poor wording. The data do not support this claim. MMSET may have many different effects and leukemia maintenance was not tested here. MMSET expression is ectopically expressed in leukemia samples compared to normal hematopoietic cells. More detail of the comparison would be helpful.

Line 149: "In a proteome-wide analysis, MMSET has been known to interact with a ubiquitin E3 ligase BRCA1 (29, Woo Park 2015). This was news to me, and the reference provided (also from these authors) does not appear support the claim. I'm sure the authors know their own work, but searching Woo Park et al paper and its supplemental information for "BRCA1" yields zero hits, and I cannot find what this is referencing. Can the authors please elaborate?"

Line 188: "DNA damage induces poly-ubiquitination of MMSET at Lys292 (ref 32, Boeing et al)." Since MMSET/WHSC1 is not mentioned in the body of the paper, this data was found by searching the on line database, and I commend the authors for finding this. The verification the authors have done is a significant advance and could be better highlighted in the abstract and/or text.

Line 246: "BRCA1 mutations induce myelogenous leukemia via dysregulation of MMSET stability" This is an overly broad claim that is not supported by the data. The conclusion of this section addresses the role of BRCA1 and MMSET on K562 cell differentiation, and does not address the induction of leukemia.

Figure 1, panel C labels should be clarified. Spleen cancer of what type? Panel f is difficult to interpret and should be removed if it can't provide better support. The panels do not look significantly different to me. Quantification of the data in Figure 1f and e is not provided and would be helpful.

Figure 2, panels b: Why does the MMSET level also decrease in Mock cells. Replate measurements are needed to perform statistical analysis and confirm a difference between Mock and hemin treated cells.

Figure 3d and elsewhere: ShRNA experiments should include control for non-expressing gene, it's not clear what the control is here. Heavy use is made of shRNA to knock down gene expression.

Figure 4 b and d: densitometry should be included here.

Figure 5: Not clear if the mRNAs studied in panels c, d, e, and f are the same genes found on unsupervised analysis in panel a. Panel g: without quantification, these data are not clear.

The authors have done considerable work on the REIIBP isoform of NSD2/MMSET. Does REIIBP have the same effect on differentiation of K562?

Line 252 "compare" should be "compared"

Line 275 "form" should be "forms"

We thank the reviewers for their insightful suggestions. Please find below our detailed responses to their individual comments.

Reviewer #1

Major revisions:

1. *In the performed genome-wide CRISPR screen, the authors only sorted cells that were treated with hemin and compared the sorted pool (CD235A+ cells) to the unsorted vehicle-treated pool. Using this strategy, it is not feasible to distinguish between genes whose loss by itself is sufficient to drive erythroid differentiation and those specifically involved in hemin-induced erythroid differentiation. The former set of genes can be identified by sorting vehicle-treated cells and identifying sgRNAs that are enriched in this population (the disruption of certain genes could result in CD235A+ cells even in the absence of hemin).*

Response: As you recommended, we tested whether several genes discovered in CRISPR screening regulate CD235A expression levels (Fig. S1C). Most genes did not affect CD235A expression levels. Besides, we confirmed CRISPR screening using RT-PCR for globin genes, o-dianisidine staining and immunoblot for GATA1 (Fig. 1C, D, and S1D). These results suggest that SOX4, PAX1, MMSET and SIRT1 accelerated hemin-mediated erythroid differentiation.

2. *Why are cells treated with hydroxide- what kind of control is this- what concentration of NaOH? Line 101- what does “innate differentiation-associated factor” mean?*

Response: The erythroid differentiation inducer, hemin can be dissolved in DMSO or NaOH. Initially, we made hemin solutions diluted in DMSO and used DMSO as a negative control. Although hemin appeared to be more efficient than DMSO in differentiation of K562 cells, however, DMSO slightly induced K562 cell differentiation. These were consistent with previous studies^{1,2}. In this study, we used NaOH (working concent. 1.2 uM) as a solvent. We removed “innate differentiation-associated factor” and changed the sentence.

3. *What do you mean by screening in the pan-cancer atlas of leukemia on line 102- reference 27 is a database of post-translational modifications of proteins and has nothing to do with leukemia.*

Response: We removed this sentence because of the discrepancy between text and the reference as you suggested.

4. *Figure 1C is uninterpretable what does multiple splenic cancers mean. Similarly what tissues does figure S1C refer to. S1D shows high level expression of MMSET in lymphoma specimens-why is this data important and how does it support a role of MMSET in lymphoma or leukemia. The topic of this manuscript is erythroid differentiation.*

Response: We understand your point that the main topic of the study is erythroid differentiation. We removed Fig. 1C multiple spleen cancer data and re-performed MMSET expression in myeloma tumors (Fig S1F). Accumulation of abnormal differentiation drives

several type of cancers³. We hypothesized overexpression of MMSET in lymphoma or myeloma was the result of accumulation of abnormal erythroid differentiation.

5-1. *In Results, line 110, the authors stated that “hemin-mediated differentiation decreased in MMSET-overexpressed K562 cells”. However, the figures (Figure 1 D-F) supporting this statement are not very convincing. In Figure 1D, Hemin treatment seems to induce CD36 expression levels in MMSET overexpressing cells, but these levels are downregulated by MMSET overexpression even in non-treated controls.*

Response: As you suggested, we measured globin gene expression instead of CD36 level for identifying K562 cell differentiation. We changed the sentence to “overexpression of MMSET disrupted hemin-mediated increase of *HBE1* and *HBA1/2* levels”.

5-2. *In figure 1E, the images have different background contrasts and the differences between o-dianisidine positive cells are not clear. Similarly, in figure 1F, the % of CD235A+ cells are not shown, and the differences are not clear. Quantification of the flow data from 3 independent experiments with statistics is necessary in this case. Alternatively, immunoblot for CD235A would clearly show the differences. The authors can also use other markers of erythroid differentiation such as globin genes (*HBA* and *HBG*) to support their hypothesis.*

Response: We change the Figure 1E and added quantification data in Fig 1D. As recommended, we evaluated erythroid differentiation markers such as globin genes (*HBE* and *HBA1/2*) (Fig. 1C) and *GATA1* expression (Fig. S1D). These results suggested that *SOX4*, *SIRT3*, *PAX1* and *MMSET* discovered in CRISPR screening affected hemin-mediated K562 differentiation. Especially, we identified overexpression of *MMSET* disrupted hemin-mediated differentiation via analyzing globin gene expressions (Fig. 1E).

5-3. *The manuscript could also benefit from using other erythroid differentiation inducers such as sodium butyrate. In the supple figure 2e shRNA knockdown of MMSET appeared to lead to differentiation even without hemin and accentuated hemin differentiation- this effect needs to be quantified.*

Response: As suggested, we performed differentiation study with sodium butyrate (Fig. S3A-C). Consistent result was obtained with hemin-treatment. In the Fig. S2E shRNA knockdown of *MMSET* appeared to lead to differentiation without hemin. Since DMSO slightly induced K562 differentiation, phenotype of K562 cell differentiation appeared in *MMSET* knockdown K562 cells. We reconfirmed Supple Fig. 2E with hemin and quantified data (Fig. S2E).

6. *Figure 2- how many times was the cycloheximide chase repeated- can an estimate of the half-life of MMSET in the presence or absence of hemin be generated. Does other differentiation inducers cause this degradation as well? MMSET has a long and short isoform- only one form is shown- no MW markers are given-is the short form seen in 293 or K562 cells?*

Response: We repeated *MMSET* half-life experiment three times and quantified them (Fig. 2C). Figure 1 (bottom) showed that half-life of *MMSET* was 9.91 hours in control which was

longer than that of hemin treatment (5.7 hours). As we describe above, sodium butyrate also caused degradation (Fig. S3A-C). In addition, we are mainly looking at long isoform of MMSET (MMSET II) in this study. Furthermore, short isoform of MMSET (MMSET I) also decreased during K562 cell differentiation (Figure 2). The result is added in the response letter below.

Figure 1. Quantification of MMSET half-life.

Figure 2. Isoforms of MMSET was reduced during K562 differentiation.

7. Line 149- reference 29 has no mention of BRCA1.

Response: Reference 29 was “RE-IIBP methylates H3K79 and induces MEIS1-mediated apoptosis via H2BK120 ubiquitination by RNF20” written by our laboratory. We identified interaction partner of MMSET isoform, RE-IIBP using LC-MS/MS analysis and notified high ranked proteins such as RNF20, BLM, RBMX, and NMNAT1. Additionally, we found BRCA1, E3 ubiquitin ligase in raw data. In previous study, BRCA1 form a complex with BLM⁴. We hypothesized MMSET might make up complex with BLM and BRCA1. We change sentence in Line 149.

8-1. *The manuscript demonstrated evidence on the physical interaction between BRCA1 and MMSET however endogenous co-precipitation was not performed and so whether the complex exists in cells is not yet shown. No complex is shown in K562 cells. The overexpression co-IP and GST pull downs are supportive but not definitive.*

Response: We performed endogenous precipitation assay using BRCA antibody and identified that interaction BRCA1 and MMSET increased during hemin-mediated K562 differentiation (Fig. 3J)

8-2. *The overexpression of BRCA1 decreased MMSET levels in 293T cells but this has to be shown in K562 cells; the knockdown of BRCA1 in 293 cells increased MMSET levels but effect in K562 are shown.*

Response: As suggested, we overexpressed BRCA1 WT or mutants in K562. Consistent with previous data, BRCA1 did not decreased MMSET levels in undifferentiated K562 cells (Figure 3, below)

Figure 3 Overexpression of BRCA1 did not affect MMSET protein level in K562 cells

8-3. *Supp Figure 3C- Basal levels of MMSET are lower in BRCA1 overexpressing cells but the change in half life has to be calculated by quantifying several blots.*

Response: We repeated MMSET half-life experiment three times (Fig. S4C). We quantified half-life of BRCA1 in BRCA1 overexpressing 293T cells. Figure 4 showed that half-life of MMSET was 46.6 hours in control cells which is longer than half-life of BRCA1 overexpressed 293T cells (10.7 hours).

Figure 4 Quantification of MMSET half-life

8-4. *Figure 3I overexpression of BRCA1 in 293 cells increased MMSET ubiquitylation- did this also happen in K562 cells. Again the results only form knockdown of BRCA1 in K562 cells is shown.*

Response: In Fig. 3H showed poly-ubiquitination level of MMSET in BRCA1 knockdown K562 cells with hemin treatment or not. Hemin treatment induced poly-ubiquitination of MMSET increased in control cells, but not in BRCA1 knockdown K562 cells (Fig. 3H). The result indicates that BRCA1 is important for degradation of MMSET during K562 differentiation.

8-5. *Line 174 is very confusing after showing that shBRCA1 prevents hemin induced decrease of MMSET levels in figure 3g- the sentence says.*

Response: We investigated the relationship between BRCA1 and MMSET in 293T/K562 cells. Fig. 3A-F showed that BRCA1 interacted with MMSET and reduced stability of MMSET via poly-ubiquitination in 293T cells. Surprisingly, we identified that knockdown of BRCA1 had no effect on endogenous MMSET expression in undifferentiated-K562 cells unlike 293T cells (Fig. 3G and H). Therefore, we hypothesized that sub-localization of BRCA1 and MMSET might be different in K562 cells.

8-6. *BRCA1 has no effect on endogenous MMSET levels. Was Fig. 3G then with overexpressed proteins or do you mean that BRCA1 knockdown does not affect MMSET levels before hemin addition- Figure 3G then would need to be repeated a few times to convincingly show this particularly since there was some increase in benzadine staining in K562 cells with shRNA BRCA1 even before hemin.*

Response: We tested a few times whether BRCA1 affect MMSET levels (Figure 5, below). Knockdown of BRCA1 did not induce MMSET level in K562 cells. We added staining of K562 cells with shRNA BRCA1 before hemin treatment (Fig. 6C). The result suggest depletion of BRCA1 didn't affect the results of o-dianisidine staining.

Figure 5 Knockdown of BRCA1 did not affect MMSET protein level.

8-7. *Supp figure 3d not very convincing co-localization since the MMSET levels are so much lower with hemin- there seems to be pretty good co-localization even before hemin. Hemin is stated to lead to transport of BRCA1 from cytoplasm to nucleus.*

Response: Following Figure 3 showed localization of BRCA1 as a full-gel image. Immunoblot result suggests that non-specific bands were mainly detected in nucleus of control cells (Figure 6A) and ICC also showed that BRCA1 seems located in both cytosol and nucleus in control cells (Figure 6B). However, we initially thought BRCA1 WT translocated from cytosol to nucleus and that's why BRCA1 was not located in cytosol in hemin-treated K562 cells in ICC. Since the data in Fig. S3D was not convincing as suggested, we deleted ICC assay result.

Figure 6 Localization of BRCA1 before and after hemin treatment.

8-8. *The IF in supp 3d shows fairly prominent BRCA1 expression in nucleus and cytoplasm while the fractionation in 3i shows almost no nuclear BRCA1- there is no cytoplasmic loading control and was this done by loading equal fractions on the gel or equal ug of protein?*

Response: We loaded 10 μ g protein for detecting cytosol and nuclear proteins. Since tubulin is a cytoplasmic marker, we used tubulin as a loading control

8-9. *3J is with overexpressed MMSET and the co-IP with BRCA1 is not convincing- the experiment should be performed with endogenous co-IP. Line 181 "endogenous interaction" is no accurate.*

Response: As suggested, we tested endogenous interaction between MMSET and BRCA1 during K562 differentiation (Fig. 3J)

9. Line 188- reference 32 does not mention MMSET ubiquityltion but there is a database resulting from this paper that indicates the polyubiquityltion. That being said figure 4 has some of the strongest data in the paper indicating that the K292 residue has a key role in stability of MMSET.

Response: Thanks for your generous comments. We identified ubiquitination of MMSET at K292 using proteomic database. We highlighted poly-ubiquitination site of MMSET in abstract and text as suggested.

10. In Figure 5, the authors selected a set of genes related to erythroid differentiation to confirm their induction by MMSET knockdown. However, key genes in erythroid differentiation including master regulators (GATA1/2), the transferrin receptor (TFRC) as well as globin genes were not addressed. Studying the regulation of such genes by MMSET would strongly support its role in erythroid differentiation. To show MMSET is specifically recruited to promoters- control PCR for regions 5', 3' to the promoter and control gene desert regions need to be performed.

Response: As suggested, we measured GATA1 and globin gene expressions. Previous studies suggested that GATA1 increased in early stage of K562 differentiation, and decreased in final stage⁵. We confirmed GATA1 decreased in Nabu- or hemin-treated K562 cells (Fig. S2D and S3A). Depletion of MMSET or BRCA1 regulated GATA1 expression during hemin-treatment (Fig. S2D and Figure 7, below). In addition, we tested globin gene expression in MMSET overexpressing or MMSET Y1118A overexpressing K562 cells during hemin-treatment (Fig. 1E). These results suggested that MMSET overexpression disrupted K562 differentiation to erythroid.

Figure 7 Knockdown of BRCA1 disrupted hemin-mediated K562 cell differentiation.

11. In Figure 5G, a control where cells with the mock vector treated with hemin needs to be added.

Response: Since NC was contained non-target shRNA, NC samples can be considered as controls in our experiments.

12. *Figure supp 5a, b as before what this protein atlas data refers to is unclear as is the term splenic cancers. Supp 5e- change “upnormal” to abnormal.*

Response: As you suggested, we changed Figure Supple. S5B and S5E.

13. *6H what is “mock”. None of the BRCA1 mutants accelerated (not the correct term; maybe enhanced) CD235a expression.*

Response: As you suggested, we change these sentence.

14. *The authors need to discuss the revealed interaction between BRCA1 and MMSET in the context of their roles as a tumor suppressor and an oncogene respectively.*

Response: As suggested, we described additional roles of BRCA1 and MMSET in discussion.

Minor Revisions:

1. *Line 53- in vivo MMSET only modulates H3K36 dimethylation; other sites are found in vitro but not in vivo.*

Response: As suggested, we changed the sentence in introduction.

2. *Line 58- which studies showed a role for MMSET in leukemia cell differentiation- none are quoted by the authors.*

Response: We added two quotations about role for MMSET in leukemia differentiation.

3. *In figure 6F, MMSET and BRCA1 immunoblot panels seem to be reversed.*

Response: We corrected mistakes in Fig 6F.

4. *In line 225, it is not clear why the authors chose RRAS2 promoter for MMSET and H3K36me3 ChIP-PCR analysis. MMSET creates the H3K36me2 mark- why isn't this being tested?*

Response: We chose RRAS2 promoter as a positive control for recruitment of MMSET⁶. In previous study, MMSET induces not only H3K36me2 but also H3K36me3⁷. So we tested whether MMSET directly regulates erythrocyte differentiation-related genes such as CD36 and BCL6 and used RRAS2 as a positive control. In Fig. S6 suggested that MMSET indirectly regulated differentiation-related genes. We hypothesized MMSET regulated these genes expression via inducing AURKA kinase activity⁸. We confirmed hemin-mediated differentiation regulated methylation of AURKA by MMSET and inhibition of AURKA induced K562 cell differentiation (Fig S6B-E). These results suggested that MMSET influenced K562 cell differentiation through indirect mechanisms.

5. In the “CRISPR-Cas9 Screening” methods section, line 393, the correct name of the used

CRISPR library should be used. It should be “Brunello” instead of CRISPR-v2. The authors also need to indicate that they used the lentiGuide-Puro library vector (2-vector system).

Response:

As you recommended, we change CRISPR-V2 to Brunello and additionally indicate 2-vector system.

6. In the “NGS library preparation and sequencing” methods section, the authors need to indicate what was the criteria used for hit selection or what method was used for gene ranking.

Response:

As you recommended, we added Data processing and analysis sections for indicating what was the criteria used for hit selection or what method was used for gene ranking.

7. In the “O-dianisidine methods section”, the authors described an O-dianisidine protocol that is used for Zebrafish embryos. The correct staining protocol used in this study should be added.

Response: We corrected staining protocol for the experiment.

8. The manuscript has many grammar mistakes and many sentences that need to be rephrased to enhance the overall clarity.

Response: We rephrased many sentences and corrected many grammatical mistakes in revised manuscript.

Reviewer #2

Detailed critique

1-1. Abstract: Line 18, “Multiple myeloma SET domain (MMSET) is linked to the IgH promoter by t(4;14)(p14;q32)...” It’s not clear why the authors focus on myeloma here given the upregulation of MMSET in other cancers too. The HGNC name is NSD2, which may reduce the confusion of readers.

Response: As suggested, we change the front of abstract for focusing on erythroid differentiation and we revise MMSET to NSD2 in the manuscript.

1-2. Line 26, “Proteomic data analysis revealed interaction..” the source for this analysis is not well presented. If this was another laboratory’s database, this should be made clear.

Response: Reference 29 was “RE-IIBP methylates H3K79 and induces MEIS1-mediated apoptosis via H2BK120 ubiquitination by RNF20” written by our laboratory. We identified interaction partner of MMSET isoform, RE-IIBP using LC-MS/MS analysis and notified high ranked proteins such as RNF20, BLM, RBMX, and NMNAT1. Additionally, we identified BRCA1, E3 ubiquitin ligase in proteomics data. Since BRCA1 has complex with BLM in the previous study⁴, we hypothesized MMSET may interacts with BRCA1. We changed the

sentence in Line 149.

1-3. Line 35, “..as a potential therapeutic target process in myeloma.” Again, why only myeloma if MMSET is upregulated in multiple cancers?

Response: Transcription activity of MMSET increased in multiple cancers. This might be independent of MMSET regulation by BRCA1. However, in this study, we found that protein level of MMSET was regulated by BRCA1 during myeloma cell differentiation. Therefore, we suggested that regulation of BRCA1-mediated MMSET ubiquitination could be a potential therapeutic target in multiple myeloma.

2. After presenting the results of their CRISPR screen, the authors begin the manuscript by providing data on MMSET from published cancer databases which is poorly presented and made worse by reaching for conclusions from those surveys that are not supported by the data. We know that MMSET is upregulated in a variety of cancers.

Response: The data we referred is from TCGA data. We removed the sentence to avoid the confusion. As suggested, we added new panels about confirmation of CRISPR screening for clarifying conclusions. We identified that SOX4, PAX1, MMSET and SIRT1 affected K562 cell differentiation to erythroid using CRISPR screening (Fig 1A and S1B). In addition, we confirmed CRISPR screening using globin gene, GATA expression and O-dianisidine staining (Fig. 1C, D and S1D). Since depletion of MMSET had strong effect on K562 differentiation and we were interested in HMTase activity of MMSET, we preceded additional experiments.

3. Line 101: “..innate differentiation-related factor, MMSET as a common candidate in both this screening and TCGA pan-cancer atlas database of leukemia (Krassowski M 2018). Can the authors clarify?”

Response: We removed the sentence to avoid the confusion.

4. Line 105: “..MMSET is upregulated during lymphoma.” This is poor wording. MMSET is over-expressed in lymphoma samples compared to what? Normal lymphoid tissue?

Response: As suggested we changed the sentence.

5. Line 110: “..suggesting that overexpression of MMSET is related to leukemia maintenance.” Poor wording. The data do not support this claim. MMSET may have many different effects and leukemia maintenance was not tested here. MMSET expression is ectopically expressed in leukemia samples compared to normal hematopoietic cells. More detail of the comparison would be helpful.

Response: As suggested, we change the sentence to “suggesting that MMSET is overexpressed in leukemia samples compared to normal hematopoietic cells and may influence leukemia maintenance.”

6. Line 149: *“In a proteome-wide analysis, MMSET has been known to interact with a ubiquitin E3 ligase BRCA1 (29, Woo Park 2015). This was news to me, and the reference provided (also from these authors) does not appear support the claim. I’m sure the authors know their own work, but searching Woo Park et al paper and its supplemental information for “BRCA1” yields zero hits, and I cannot find what this is referencing. Can the authors please elaborate?”*

Response: Reference 29 was “RE-IIBP methylates H3K79 and induces MEIS1-mediated apoptosis via H2BK120 ubiquitination by RNF20” written by our laboratory. We identified interaction partner of MMSET isoform, RE-IIBP using LC-MS/MS analysis and notified high ranked proteins such as RNF20, BLM, RBMX, and NMNAT1. Additionally, we identified BRCA1, E3 ubiquitin ligase in proteomics data. Since BRCA1 has complex with BLM in the previous study⁴, we hypothesized MMSET may interacts with BRCA1. We changed the sentence in Line 149.

7. Line 188: *“DNA damage induces poly-ubiquitination of MMSET at Lys292 (ref 32, Boeing et al).” Since MMSET/WHSC1 is not mentioned in the body of the paper, this data was found by searching the on line database, and I commend the authors for finding this. The verification the authors have done is a significant advance and could be better highlighted in the abstract and/or text.*

Response: We identified ubiquitination of MMSET at K292 using proteomic database. We highlighted poly-ubiquitination site of MMSET in abstract and text as suggested.

8. Line 246: *“BRCA1 mutations induce myelogenous leukemia via dysregulation of MMSET stability” This is an overly broad claim that is not supported by the data. The conclusion of this section addresses the role of BRCA1 and MMSET on K562 cell differentiation, and does not address the induction of leukemia.*

Response: As suggested, we toned down the sentence and changed it to “The role of BRCA1 and NSD2 on K562 differentiation”.

9. Fig. 1, panel C labels should be clarified. Spleen cancer of what type? Panel f is difficult to interpret and should be removed if it can’t provide better support. The panels do not look significantly different to me. Quantification of the data in Figure 1f and e is not provided and would be helpful.

Response: We removed the Fig. 1C to avoid confusion and added myeloma tissue array data (Fig. S1F) and quantified Fig. 1f. in addition, we investigated several gene such as MMSET, SOX4, PAX1 and SIRT3 related to K562 differentiation and quantified it (Fig. 1D)

10. Fig. 2, panel b: Why does the MMSET level also decrease in Mock cells. Repeat measurements are needed to perform statistical analysis and confirm a difference between Mock and hemin treated cells.

Response: We repeated the experiment (Figure 8, below) and changed Fig. 2 panel b.

Figure 8 Stability of MMSET decreased in hemin-treated K562 cells.

11. *Figure 3d and elsewhere: ShRNA experiments should include control for non-expressing gene, it's not clear what the control is here. Heavy use is made of shRNA to knock down gene expression.*

Response: Since NC was contained non-target shRNA, NC samples can be considered as controls in our experiments.

12. *Figure 4 b and d: densitometry should be included here.*

Response: we quantified the Fig. 4b and d.

13. *Figure 5: Not clear if the mRNAs studied in panels c, d, e, and f are the same genes found on unsupervised analysis in panel a. Panel g: without quantification, these data are not clear.*

Response: We magnified the clustering image and indicated where those genes are located. Since Fig. 1G and Fig 5G data were not clear, we added relative percent of CD235A positive cells via quantification (Fig. 1H).

14. *The authors have done considerable work on the REIIBP isoform of NSD2/MMSET. Does REIIBP have the same effect on differentiation of K562?*

Response: It is very hard to perform the additional experiment since no RE-IIBP antibodies are available. We previously confirmed the data which shows no effect of RE-IIBP in K562 treated with hemin (Figure 6, below). We found that MMSET K292 was target for ubiquitination, but REIIBP have no ubiquitination site. Therefore, we speculate that REIIBP might not be affected by hemin-treatment.

Figure 6 MMSET expression was decreased during K562 differentiation but REIIBP was not affected.

15. Line 252 “compare” should be “compared”.

Response: We changed it to “compared”.

16. Line 275 “form” should be “forms”.

Response: We changed it to “forms”.

We hope that you will agree that our revised manuscript provides sufficient evidences to demonstrate that BRCA1-mediated ubiquitination of MMSET induces proteosomal degradation of MMSET which is required for hemin-induced erythroid differentiation in K562 cells. We are certain that these results provide a valuable resource for further research into the post translational modifications of MMSET and introduce another important key protein BRCA1 in hematopoietic cell differentiation.

We also hope that you will find the resubmitted manuscript both improved and appropriate for Communication Biology.

REFERENCES

1. Zhou M, *et al.* Histone demethylase RBP2 decreases miR-21 in blast crisis of chronic myeloid leukemia. *Oncotarget* **6**, 1249-1261 (2015).
2. Sutherland JA, Turner AR, Mannoni P, McGann LE, Turc JM. Differentiation of K562 leukemia cells along erythroid, macrophage, and megakaryocyte lineages. *J Biol Response Mod* **5**, 250-262 (1986).
3. Nefedova Y, *et al.* Hyperactivation of STAT3 is involved in abnormal differentiation of dendritic cells in cancer. *Journal of immunology* **172**, 464-474 (2004).
4. Acharya S, *et al.* Association of BLM and BRCA1 during Telomere Maintenance in ALT Cells. *PloS one* **9**, e103819 (2014).
5. Ferreira R, Ohneda K, Yamamoto M, Philipsen S. GATA1 function, a paradigm for transcription factors in hematopoiesis. *Molecular and cellular biology* **25**, 1215-1227 (2005).

6. Kuo AJ, *et al.* NSD2 links dimethylation of histone H3 at lysine 36 to oncogenic programming. *Molecular cell* **44**, 609-620 (2011).
7. Yang P, *et al.* Histone methyltransferase NSD2/MMSET mediates constitutive NF-kappaB signaling for cancer cell proliferation, survival, and tumor growth via a feed-forward loop. *Molecular and cellular biology* **32**, 3121-3131 (2012).
8. Park JW, Chae YC, Kim JY, Oh H, Seo SB. Methylation of Aurora kinase A by MMSET reduces p53 stability and regulates cell proliferation and apoptosis. *Oncogene* **37**, 6212-6224 (2018).

Reviewers' comments:

Reviewer #1 (Remarks to the Author):

Re-review Park et al

There have been many responses to this reviewer but questions remain. My response to specific changes are below as are some additional comments

Page 4 line 14, 15 – these references do not discuss leukemia cell differentiation; I know of no data that related NSD2 to leukemia

Line 89 guides are not shRNAs

Line 93- do you mean hemin induced CD235+ cells

Reviewer #1

1. In the performed genome-wide CRISPR screen, the authors only sorted cells that were treated with hemin and compared the sorted pool (CD235A+ cells) to the unsorted vehicle-treated pool. Using this strategy, it is not feasible to distinguish between genes whose loss by itself is sufficient to drive erythroid differentiation and those specifically involved in hemin-induced erythroid differentiation. The former set of genes can be identified by sorting vehicle-treated cells and identifying sgRNAs that are enriched in this population (the disruption of certain genes could result in CD235A+ cells even in the absence of hemin).

Response: As you recommended, we tested whether several genes discovered in CRISPR screening regulate CD235A expression levels (Fig. S1C). Most genes did not affect CD235A expression levels. Besides, we confirmed CRISPR screening using RT-PCR for globin genes, o-dianisidine staining and immunoblot for GATA1 (Fig. 1C, D, and S1D). These results suggest that SOX4, PAX1, MMSET and SIRT1 accelerated hemin-mediated erythroid differentiation.

Reviewer response Supp 1D decreases in GATA 1 with guide against SOX4 are not convincing

2. Why are cells treated with hydroxide- what kind of control is this- what concentration of NaOH?
Line 101- what does "innate differentiation-associated factor" mean?

Response: The erythroid differentiation inducer, hemin can be dissolved in DMSO or NaOH. Initially, we made hemin solutions diluted in DMSO and used DMSO as a negative control. Although hemin appeared to be more efficient than DMSO in differentiation of K562 cells, however, DMSO slightly induced K562 cell differentiation. These were consistent with previous studies^{1, 2}. In this study, we used NaOH (working concent. 1.2 uM) as a solvent. We removed "innate differentiation-associated factor" and changed the sentence.

Review response- OK

3. What do you mean by screening in the pan-cancer atlas of leukemia on line 102- reference 27 is a database of post-translational modifications of proteins and has nothing to do with leukemia.

Response: We removed this sentence because of the discrepancy between text and the reference as you suggested.

Review response- OK

4. Figure 1C is uninterpretable what does multiple splenic cancers mean. Similarly what tissues does figure S1C refer to. S1D shows high level expression of MMSET in lymphoma specimens-why is this data important and how does it support a role of MMSET in lymphoma or leukemia. The topic of this manuscript is erythroid differentiation.

Response: We understand your point that the main topic of the study is erythroid differentiation. We removed Fig. 1C multiple spleen cancer data and re-performed MMSET expression in myeloma tumors (Fig S1F). Accumulation of abnormal differentiation drives several type of cancers³. We hypothesized overexpression of MMSET in lymphoma or myeloma was the result of accumulation of abnormal erythroid differentiation.

Reviewer response- all of these data are irrelevant to the paper and should be eliminate (supp fig 1 e and f); eliminate text lines 106-112

5-1. In Results, line 110, the authors stated that "hemin-mediated differentiation decreased in MMSET-overexpressed K562 cells". However, the figures (Figure 1 D-F) supporting this statement are not very convincing. In Figure 1D, Hemin treatment seems to induce CD36 expression levels in MMSET overexpressing cells, but these levels are downregulated by MMSET overexpression even in non-treated controls.

Response: As you suggested, we measured globin gene expression instead of CD36 level for identifying K562 cell differentiation. We changed the sentence to "overexpression of MMSET disrupted hemin-mediated increase of HBE1 and HBA1/2 levels".

5-2. In figure 1E, the images have different background contrasts and the differences between o-dianisidine positive cells are not clear. Similarly, in figure 1F, the % of CD235A+ cells are not shown, and the differences are not clear. Quantification of the flow data from 3 independent experiments with statistics is necessary in this case. Alternatively, immunoblot for CD235A would clearly show the differences. The authors can also use other markers of erythroid differentiation such as globin genes (HBA and HBG) to support their hypothesis.

Review response- Figure 1d remains unconvincing. 1H needs quantification of MFI not percent positive- where are the positive versus negative gates drawn?

Response: We change the Figure 1E and added quantification data in Fig 1D. As recommended, we evaluated erythroid differentiation markers such as globin genes (HBE and HBA1/2) (Fig. 1C) and GATA1 expression (Fig. S1D). These results suggested that SOX4, SIRT3, PAX1 and MMSET discovered in CRISPR screening affected hemin-mediated K562 differentiation. Especially, we identified overexpression of MMSET disrupted hemin-mediated differentiation via analyzing globin gene expressions (Fig. 1E).

5-3. The manuscript could also benefit from using other erythroid differentiation inducers such as sodium butyrate. In the supple figure 2e shRNA knockdown of MMSET appeared to lead to differentiation even without hemin and accentuated hemin differentiation- this effect needs to be quantified.

Response: As suggested, we performed differentiation study with sodium butyrate (Fig. S3A-C). Consistent result was obtained with hemin-treatment. In the Fig. S2E shRNA knockdown of MMSET appeared to lead to differentiation without hemin. Since DMSO slightly induced K562

differentiation, phenotype of K562 cell differentiation appeared in MMSET knockdown K562 cells. We reconfirmed Supple Fig. 2E with hemin and quantified data (Fig. S2E).

Review response- Supp figure 2f – how can you have a fold increase over 0?

Line 142- How do you know K562 differentiation through NaBu is due to decreased GATA1? Gata1 is required for erythroid cells.

6. Figure 2- how many times was the cycloheximide chase repeated- can an estimate of the half-life of MMSET in the presence or absence of hemin be generated. Does other differentiation inducers cause this degradation as well? MMSET has a long and short isoform- only one form is shown- no MW markers are given-is the short form seen in 293 or K562 cells?

Response: We repeated MMSET half-life experiment three times and quantified them (Fig. 2C). Figure 1 (bottom) showed that half-life of MMSET was 9.91 hours in control which was longer than that of hemin treatment (5.7 hours). As we describe above, sodium butyrate also caused degradation (Fig. S3A-C). In addition, we are mainly looking at long isoform of MMSET (MMSET II) in this study. Furthermore, short isoform of MMSET (MMSET I) also decreased during K562 cell differentiation (Figure 2). The result is added in the response letter below.

Figure 1. Quantification of MMSET half-life.

Figure 2. Isoforms of MMSET was reduced during K562 differentiation.

7. Line 149- reference 29 has no mention of BRCA1.

Response: Reference 29 was "RE-IIBP methylates H3K79 and induces MEIS1-mediated apoptosis via H2BK120 ubiquitination by RNF20" written by our laboratory. We identified interaction partner of MMSET isoform, RE-IIBP using LC-MS/MS analysis and notified high ranked proteins such as RNF20, BLM, RBMX, and NMNAT1. Additionally, we found BRCA1, E3 ubiquitin ligase in raw data. In previous study, BRCA1 form a complex with BLM4. We hypothesized MMSET might make up complex with BLM and BRCA1. We change sentence in Line 149.

Review response- Re-word line 154- maybe "From data obtained in a prior study...."

8-1. The manuscript demonstrated evidence on the physical interaction between BRCA1 and MMSET however endogenous co-precipitation was not performed and so whether the complex exists in cells is not yet shown. No complex is shown in K562 cells. The overexpression co-IP and GST pull downs are supportive but not definitive.

Response: We performed endogenous precipitation assay using BRCA antibody and identified that interaction BRCA1 and MMSET increased during hemin-mediated K562 differentiation (Fig. 3J)

8-2. The overexpression of BRCA1 decreased MMSET levels in 293T cells but this has to be shown in K562 cells; the knockdown of BRCA1 in 293 cells increased MMSET levels but effect in K562 are shown.

Response: As suggested, we overexpressed BRCA1 WT or mutants in K562. Consistent with previous data, BRCA1 did not decreased MMSET levels in undifferentiated K562 cells (Figure 3,

below)

Figure 3 Overexpression of BRCA1 did not affect MMSET protein level in K562 cells

8-3. Supp Figure 3C- Basal levels of MMSET are lower in BRCA1 overexpressing cells but the change in half life has to be calculated by quantifying several blots.

Review response- What happened to NSD2 stability with overexpression of BRCA1 in K562 and induction of differentiation?

Response: We repeated MMSET half-life experiment three times (Fig. S4C). We quantified half-life of BRCA1 in BRCA1 overexpressing 293T cells. Figure 4 showed that half-life of MMSET was 46.6 hours in control cells which is longer than half-life of BRCA1 overexpressed 293T cells (10.7 hours).

Figure 4 Quantification of MMSET half-life

8-4. Figure 3I overexpression of BRCA1 in 293 cells increased MMSET ubiquitylation- did this also happen in K562 cells. Again the results only form knockdown of BRCA1 in K562 cells is shown.

Response: In Fig. 3H showed poly-ubiquitination level of MMSET in BRCA1 knockdown K562 cells with hemin treatment or not. Hemin treatment induced poly-ubiquitination of MMSET increased in control cells, but not in BRCA1 knockdown K562 cells (Fig. 3H). The result indicates that BRCA1 is important for degradation of MMSET during K562 differentiation.

8-5. Line 174 is very confusing after showing that shBRCA1 prevents hemin induced decrease of MMSET levels in figure 3g- the sentence says.

Response: We investigated the relationship between BRCA1 and MMSET in 293T/K562 cells. Fig. 3A-F showed that BRCA1 interacted with MMSET and reduced stability of MMSET via poly-ubiquitination in 293T cells. Surprisingly, we identified that knockdown of BRCA1 had no effect on endogenous MMSET expression in undifferentiated-K562 cells unlike 293T cells (Fig. 3G and H). Therefore, we hypothesized that sub-localization of BRCA1 and MMSET might be different in K562 cells.

8-6. BRCA1 has no effect on endogenous MMSET levels. Was Fig. 3G then with overexpressed proteins or do you mean that BRCA1 knockdown does not affect MMSET levels before hemin addition- Figure 3G then would need to be repeated a few times to convincingly show this particularly since there was some increase in benzadine staining in K562 cells with shRNA BRCA1 even before hemin.

Response: We tested a few times whether BRCA1 affect MMSET levels (Figure 5, below). Knockdown of BRCA1 did not induce MMSET level in K562 cells. We added staining of K562 cells with shRNA BRCA1 before hemin treatment (Fig. 6C). The result suggest depletion of BRCA1 didn't affect the results of o-dianisidine staining.

Figure 5 Knockdown of BRCA1 did not affect MMSET protein level.

8-7. Supp figure 3d not very convincing co-localization since the MMSET levels are so much lower with hemin- there seems to be pretty good co-localization even before hemin. Hemin is stated to

lead to transport of BRCA1 from cytoplasm to nucleus.

Response: Following Figure 3 showed localization of BRCA1 as a full-gel image. Immunoblot result suggests that non-specific bands were mainly detected in nucleus of control cells (Figure 6A) and ICC also showed that BRCA1 seems located in both cytosol and nucleus in control cells (Figure 6B). However, we initially thought BRCA1 WT translocated from cytosol to nucleus and that's why BRCA1 was not located in cytosol in hemin-treated K562 cells in ICC. Since the data in Fig. S3D was not convincing as suggested, we deleted ICC assay result.

Figure 6 Localization of BRCA1 before and after hemin treatment.

8-8. The IF in supp 3d shows fairly prominent BRCA1 expression in nucleus and cytoplasm while the fractionation in 3i shows almost no nuclear BRCA1- there is no cytoplasmic loading control and was this done by loading equal fractions on the gel or equal ug of protein?

Response: We loaded 10 μ g protein for detecting cytosol and nuclear proteins. Since tubulin is a cytoplasmic marker, we used tubulin as a loading control

8-9. 3J is with overexpressed MMSET and the co-IP with BRCA1 is not convincing- the experiment should be performed with endogenous co-IP. Line 181 "endogenous interaction" is no accurate.

Response: As suggested, we tested endogenous interaction between MMSET and BRCA1 during K562 differentiation (Fig. 3J)

Review response- OK

9. Line 188- reference 32 does not mention MMSET ubiquityltion but there is a database resulting from this paper that indicates the polyubiquityltion. That being said figure 4 has some of the strongest data in the paper indicating that the K292 residue has a key role in stability of MMSET.

Response: Thanks for your generous comments. We identified ubiquitination of MMSET at K292 using proteomic database. We highlighted poly-ubiquitination site of MMSET in abstract and text as suggested.

10. In Figure 5, the authors selected a set of genes related to erythroid differentiation to confirm their induction by MMSET knockdown. However, key genes in erythroid differentiation including master regulators (GATA1/2), the transferrin receptor (TFRC) as well as globin genes were not addressed. Studying the regulation of such genes by MMSET would strongly support its role in erythroid differentiation. To show MMSET is specifically recruited to promoters- control PCR for regions 5', 3' to the promoter and control gene desert regions need to be performed.

Response: As suggested, we measured GATA1 and globin gene expressions. Previous studies suggested that GATA1 increased in early stage of K562 differentiation, and decreased in final stage5. We confirmed GATA1 decreased in Nabu- or hemin-treated K562 cells (Fig. S2D and S3A). Depletion of MMSET or BRCA1 regulated GATA1 expression during hemin-treatment (Fig. S2D and Figure 7, below). In addition, we tested globin gene expression in MMSET overexpressing or MMSET Y1118A overexpressing K562 cells during hemin-treatment (Fig. 1E). These results suggested that MMSET overexpression disrupted K562 differentiation to erythroid.

Figure 7 Knockdown of BRCA1 disrupted hemin-mediated K562 cell differentiation.

Review response- ChIP experiments still do not have controls of 5', 3' to promoter and gene desert

11. In Figure 5G, a control where cells with the mock vector treated with hemin needs to be added.

Response: Since NC was contained non-target shRNA, NC samples can be considered as controls in our experiments.

12. Figure supp 5a, b as before what this protein atlas data refers to is unclear as is the term splenic cancers. Supp 5e- change "upnormal" to abnormal.

Response: As you suggested, we changed Figure Supple. S5B and S5E.

13. 6H what is "mock". None of the BRCA1 mutants accelerated (not the correct term; maybe enhanced) CD235a expression.

Response: As you suggested, we change these sentence.

14. The authors need to discuss the revealed interaction between BRCA1 and MMSET in the context of their roles as a tumor suppressor and an oncogene respectively.

Response: As suggested, we described additional roles of BRCA1 and MMSET in discussion.

Minor Revisions:

1. Line 53- in vivo MMSET only modulates H3K36 dimethylation; other sites are found in vitro but not in vivo.

Response: As suggested, we changed the sentence in introduction.

2. Line 58- which studies showed a role for MMSET in leukemia cell differentiation- none are quoted by the authors.

Response: We added two quotations about role for MMSET in leukemia differentiation.

3. In figure 6F, MMSET and BRCA1 immunoblot panels seem to be reversed.

Response: We corrected mistakes in Fig 6F.

4. In line 225, it is not clear why the authors chose RRAS2 promoter for MMSET and H3K36me3 ChIP-PCR analysis. MMSET creates the H3K36me2 mark- why isn't this being tested?

Response: We chose RRAS2 promoter as a positive control for recruitment of MMSET6. In previous study, MMSET induces not only H3K36me2 but also H3K36me37. So we tested whether MMSET directly regulates erythrocyte differentiation-related genes such as CD36 and BCL6 and used RRAS2 as a positive control. In Fig. S6 suggested that MMSET indirectly regulated differentiation-related genes. We hypothesized MMSET regulated these genes expression via inducing AURKA kinase activity8. We confirmed hemin-mediated differentiation regulated methylation of AURKA by MMSET and inhibition of AURKA induced K562 cell differentiation (Fig S6B-E). These results suggested that MMSET influenced K562 cell differentiation through indirect mechanisms.

Review response- NSD2 does not cause H3K36me3, its recruitment is increased with transcriptional elongation and H3K36me3 but that's indirect through SETD2

5. In the "CRISPR-Cas9 Screening" methods section, line 393, the correct name of the used CRISPR library should be used. It should be "Brunello" instead of CRISPR-v2. The authors also need to indicate that they used the lentiGuide-Puro library vector (2-vector system).

Response:

As you recommended, we change CRISPR-V2 to Brunello and additionally indicate 2-vector system.

6. In the "NGS library preparation and sequencing" methods section, the authors need to indicate what was the criteria used for hit selection or what method was used for gene ranking.

Response:

As you recommended, we added Data processing and analysis sections for indicating what was the criteria used for hit selection or what method was used for gene ranking.

7. In the "O-dianisidine methods section", the authors described an O-dianisidine protocol that is used for Zebrafish embryos. The correct staining protocol used in this study should be added.

Response: We corrected staining protocol for the experiment.

8. The manuscript has many grammar mistakes and many sentences that need to be rephrased to enhance the overall clarity.

Response: We rephrased many sentences and corrected many grammatical mistakes in revised manuscript.

Reviewer #2 (Remarks to the Author):

Park et al have responded robustly to my critiques and that of the other reviewer. The data are strong and support a role of NSD2 in hematopoietic cell differentiation. The biochemical mechanism for control of NSD2 protein levels is a valuable contribution and will advance this field. I recommend publication. Strong work!

Reviewer 1

Major revisions:

There have been many responses to this reviewer but questions remain. My response to specific changes are below as are some additional comments

1. *Reviewer response Supp 1D decreases in GATA 1 with guide against SOX4 are not convincing*

Response : As suggested, we repeated the western blot assay, and revised Figure S1D.

2. *Reviewer response- all of these data are irrelevant to the paper and should be eliminate (supp fig 1 e and f); eliminate text lines 106-112*

Response :As suggested, we removed these figures.

3. *Review response- Figure 1d remains unconvincing. 1H needs quantification of MFI not percent positive- where are the positive versus negative gates drawn?*

Response: We change the Figure 1D and added the positive vs negative gates drawn in Figure 1G and Figure 5G. In addition, we revised quantification of Figure 1H

4. *Review response- Supp figure 2f – how can you have a fold increase over 0?*

Response: We showed number of o-dianisidine positive cells compare to that of negative cells. Because o-dianisidine positive cell was not detected in hemin untreated cells, we represented '0' in shNC or shNSD2.

5. *Review response- Re-word line 154- maybe “From data obtained in a prior study....*

Response: As suggested, we changed the sentence.

6. *Review response- What happened to NSD2 stability with overexpression of BRCA1 in K562 and induction of differentiation?*

Response: We confirmed BRCA1 interacted with NSD2 and enhanced ubiquitination of NSD2 in Figure 3. We hypothesized BRCA1 reduced NSD2 stability through the direct or indirect mechanism. Previous study suggested that BRCA1 was mainly localized in nucleus¹. However, in case of K562 cell, we identified that BRCA1 is localized in cytosol. Erythroid differentiation induces translocation of BRCA1 to nucleus and interaction between BRCA1 and NSD2 followed by reducing NSD2 stability. Unlike 293T cell, knockdown of BRCA1 did not affect NSD2 protein stability in K562 cells.

7. *Review response- ChIP experiments still do not have controls of 5', 3' to promoter and gene desert*

Response: As suggested, we added distal and proximal region of promoter and gene body region of each genes including BCL6, CD36, HBE1, and RRAS2 in Figure S6A. Consistent with our previous data, NSD2 did not directly regulate differentiation-related genes.

Minor Revisions:

1. *Page 4 line 14, 15 – these references do not discuss leukemia cell differentiation; I know of no data that related NSD2 to leukemia*

Response: We meant that these references are about CRISPR/Cas9 screening we used. Because it can be confusing, we removed these references.

2. *Line 89 guides are not shRNAs*

Response: We changed shRNAs to sgRNAs

3. *Line 93- do you mean hemin induced CD235+ cells*

Response: We performed CRISPR/Cas9 screening using CD235A as a marker of erythroid differentiation for understanding differentiation mechanisms. We identified several genes such as PAX1, SIRT3, SOX4, and NSD2. Figure S1C were conducted to determine whether these genes are the result of false positives of the screening by regulating the transcription of CD235A. Since these genes did not regulated CD235A transcription, we hypothesized that these gene affects K562 cell differentiation.

4. *Review response- NSD2 does not cause H3K36me3, its recruitment is increased with transcriptional elongation and H3K36me3 but that's indirect through SETD2*

Response: As suggested, we conducted ChIP assay using H3K36me2 in promoter and gene-body regions of each genes (Figure S6A).

We hope that you will agree that our revised manuscript provides sufficient evidences to demonstrate that BRCA1-mediated ubiquitination of MMSET induces proteosomal degradation of MMSET which is required for hemin-induced erythroid differentiation in K562 cells. We are certain that these results provide a valuable resource for further research into the post translational modifications of MMSET and introduce another important key protein BRCA1 in hematopoietic cell differentiation.

We also hope that you will find the resubmitted manuscript both improved and appropriate for Communications Biology.

REFERENCES

1. Feng L, *et al.* Cell cycle-dependent inhibition of 53BP1 signaling by BRCA1. *Cell Discov* **1**, 15019 (2015).

REVIEWERS' COMMENTS:

Reviewer #1 (Remarks to the Author):

The authors have made many corrections to the manuscript as suggested and there are a few minor issues remaining

Page 3 line 54- ref 14 and 15 are not about leukemia differentiation and NSD2 - this was pointed out in the last review.

1- Figure 1d-right - typo CRISPR not CRIPR

Figure 1f- if you are stating that Y1118A causes some increase in staining- please indicate with a bar graph and whether its significant. see line 113 text as well

Figure 1g- Its still difficult to tel the effects of wt NSD2 versus Y1118A without shoing there percent positive in a gate and the men fluorescent intensity. also line 118 text

Figure 2C. Quantification of the blots shows that the t/12 of the NSD2 protein in mock is > 8 hours and with Hemin 4 hours. not the 59 and 24 minutes. steted in the text- please reconcile.

We thank the reviewer and editors for their helpful suggestions.
Please find below our responses to each comment.

Reviewer #1

1. *Page 3 line 54- ref 14 and 15 are not about leukemia differentiation and NSD2*

Response: We changed the sentence from leukemia differentiation to hematopoietic differentiation. We removed the reference 15.

2. *Figure 1d-right - typo CRISPR not CRIPR.*

Response: We corrected the typo.

3. *Figure 1f- if you are stating that Y1118A causes some increase in staining- please indicate with a bar graph and whether its significant. see line 113 text as well.*

Response: We added the bar graph to Supplementary Figure 1E.

4. *Figure 1g- Its still difficult to tel the effects of wt NSD2 versus Y1118A without shooing there percent positive in a gate and the men fluorescent intensity. also line 118 text*

Response: We compared florescent positive Y1118A to control cells and added statistical significance.

5. *Figure 2C. Quantification of the blots shows that the t/12 of the NSD2 protein in mock is > 8 hours and with Hemin 4 hours. not the 59 and 24 minutes. stated in the text- please reconcile.*

Response: We changed sentence to 8 h and 4 h, respectively.

Thank you very much and we look forward to hearing from you shortly.